

# A 1-km daily soil moisture dataset of China based on in-situ measurement using machine learning

Qingliang Li[1,2], Gaosong Shi[2], Wei Shangguan [1, *], Jianduo Li[3], Lu Li[1], Feini Huang[1], Ye Zhang[1], Chunyan Wang[2] Dagang Wang[4], Jianxiu Qiu[4], Xingjie Lu[1], Yongjiu Dai[1]

[1]Southern Marine Science and Engineering Guangdong Laboratory (Zhuhai), Guangdong Province Key Laboratory for Climate Change and Natural Disaster Studies, School of Atmospheric Sciences, Sun Yat-sen University, Guangzhou 510275, China;
[2]College of Computer Science and Technology, Changchun Normal University, Changchun 130032, China;
[3]State Key Laboratory of Severe Weather, Chinese Academy of Meteorological Sciences, Beijing 10081, China
[4]School of Geography and Planning, Sun Yat-sen University, Guangzhou 510275, China

*Correspondence to*: Wei Shangguan (Email: shgwei@mail.sysu.edu.cn)

**Abstract.** High quality gridded soil moisture products are essential for many Earth system science applications, and they are usually available from remote sensing or model simulations with coarse resolution. Here we present a 1 km resolution long-term dataset of soil moisture derived through machine learning trained with *in-situ* measurements of 1,789 stations, named as SMCI1.0. Random Forest is used to predict soil moisture using ERA5-land time series, leaf area index, land cover type, topography and soil properties as covariates. SMCI1.0 provides 10-layer soil moisture with 10 cm intervals up to 100 cm deep at daily resolution over the period 2010-2020. Using *in-situ* soil moisture as the benchmark, two independent experiments are conducted to investigate the estimation accuracy of the SMCI1.0: year-to-year experiment (*ubRMSE* ranges from 0.041-0.052 and R ranges from 0.883-0.919) and station-to-station experiment (*ubRMSE* ranges from 0.045-0.051 and R ranges from 0.866-0.893). SMCI1.0 generally has advantages over other gridded soil moisture products, including ERA5-Land, SMAP-L4 and SoMo.ml. However, the high errors of soil moisture often located in North China Monsoon Region. Overall, the highly accurate estimations of both the year-to-year and station-to-station experiments ensure the applicability of SMCI1.0 to studies on the spatial-temporal patterns. As SMCI1.0 is based on *in-situ* data, it can be useful complements of existing model-based and satellite-based datasets for various hydrological, meteorological, and ecological analyses and modeling. The DOI link for the dataset is http://dx.doi.org/10.11888/Terre.tpdc.272415 (Shangguan et al., 2022).

## 1 Introduction

Soil moisture (SM) plays a key role in land-atmosphere interactions through its strong impacts on water and carbon cycle (Entekhabi et al. 1996; Seneviratne et al. 2010; Wagner et al. 2007). The status of SM is closely related to the variation in climate and weather (Dirmeyer et al. 2006). The high-quality SM with large spatial-temporal scale can be valued as indispensable factors for observing the extreme weather events, e.g. monitoring of droughts (Chawla et al. 2020; Mishra et al. 2017; Tijdeman and Menzel 2021) and floods (Kim et al. 2019; Norbiato et al. 2008; Parinussa et al. 2016). Hence, high-quality SM can be acted as a vital variable in wide range of applications such as flood and drought prediction and carbon cycle modelling (Sungmin and Orth 2020). Further, SM is also identified as an important component of the Essential Climate Variables by the Global Observing System for Climate (GCOS 2016). However, high-quality SM data acquisition is a challenging task due to the high variability of SM in space and time (Li and Lin 2018; Ojha et al. 2014; Vereecken et al. 2014). The variations in SM are affected by the inherent heterogeneity of soils, land cover, and weather (Brocca et al. 2007; Crow et al. 2012; Vereecken et al. 2014).



At present, the way of SM data acquisition can be divided into five categories: *in-situ* SM stations, satellite observations, offline land surface model simulations, Earth system model simulations, and reanalysis products. For *in-situ* SM observations, SM data is usually measured by probe measurement method (Orth and Seneviratne 2014), they have lower errors than satellite observations, land surface model simulations, Earth system model simulations and reanalysis products (Pan et al. 2019). Although large number of stations have distributed all over the world, there are still many regions with no *in-situ* SM observations due to financial constraints (Karthikeyan and Kumar 2016) and they are too sparse to capture adequate spatial coverage (Gruber et al. 2016). For satellite observations, SM data is mainly retrieved by microwave radiometer (frequencies are less than 12 GHz) on satellite (Entekhabi et al. 2010; Fujii et al. 2009; Kerr and Coauthors 2010) which can provide the global SM data with uniformly distribution. But for the microwave radiometer measured SM data from the near-surface, only the top layer SM (typically ~5 cm) can be retrieved and the data gaps exist in regions with dense vegetation, and snow-covered or frozen soils. The SM in offline land surface model and Earth system model simulations spans multiple soil layers and have seamless spatial distribution (Gu et al. 2019), but they both have the uncertain and different forcing factors due to the spatial sub-grid heterogeneity of soil properties and vegetation, thus leading to large differences from *in-situ* SM observations. (Dirmeyer et al. 2006; Kumar et al. 2009). For reanalysis products, they can also provide SM data with well temporal variations by assimilating observations into land surface models or Earth system model (Chen et al. 2021). Meanwhile, they can also provide SM data in deeper soil depth than satellite observations. However, reanalysis products still have the differences with *in-situ* SM observations when the assimilated meteorological variables (e.g., precipitation) are biased (Balsamo et al. 2015).

In brief, the characteristic strong-points and shortcomings are both coexisted in each type of SM product. Hence, we are eager to develop the high-quality SM product which comprehensively have high-resolution seamless spatial distribution, long time periods, and low errors from the above SM products.

Recently, machine-learning (ML) models have been successfully applied in SM prediction (Li et al. 2021; Mohamed et al. 2021; Xu et al. 2010) and downscale modeling (Chakrabarti et al. 2014; Srivastava et al. 2013; Wei et al. 2019). They capture the complex nonlinear relationship between SM and all available predictors related to SM variation (e.g., meteorological variables, land-cover and soil data) and further achieve accurate results. ML models provide an alternative opportunity for estimating high-quality SM data based on *in-situ* SM stations (Sungmin and Orth 2020) and further improve the generated SM product, that give full play to the roles of the *in-situ* SM observations with low errors, and other SM products with seamless spatial distribution and long time periods. Such as, Zeng et al. (Zeng et al. 2019) applied the random forest (RF) model to generate 0.5 km spatial and daily temporal resolution of SM observations over the period from 2010 to 2014 in Oklahoma based on *in-situ* SM stations and satellite observations. The low root means square error (ranging from 0.038 to 0.050 $m^3/m^3$ for year-to-year test and 0.044 to 0.057 for station-to-station test) obtained from experiments, which demonstrated the usability of their SM data. Sungmin et al. (Sungmin and Orth 2020) used the Long Short-term Memory (LSTM) model to estimate SM data in the whole word with about 27.75 km spatial and daily temporal resolution over the period from 2000 to 2019. They represented that their SM data outperformed the SM datasets of ERA5. It was necessary to note that the above two studies both emphasized that the applied *in-situ* SM observations did not cover the whole tested regions, leading to relatively high uncertainty outside the training conditions. In other words, the more *in-situ* SM stations existed in the tested region, the high-quality gridded SM data can be generated by ML models. Additionally, Carranza et al. (Carranza et al. 2020) used RF model to estimate root zone SM within a small catchment from 2016 to 2018, and demonstrated that ML model had slightly higher accuracy than a process-based model combined with data assimilation for data-poor regions. Karthikeyan et al. (Karthikeyan and Mishra 2021) applied Extreme Gradient Boosting (XGBoost) to estimate SM data in the United States with about 1 km spatial and daily temporal resolution over the period from 31 March



2015 to 29 February 2019 (only 1431 days) and the results showed that they can well capture temporal variations of SM ($ubRMSE$ less than 0.04 m³/m³).

China is one of the largest countries in the world, which located central and eastern Asia. The climate types are complex and diverse, which spans wet, semi humid, semi dry and dry climate types from southeast to northwest, the northward extent and intensity of summer monsoon often cause significant changes in precipitation and arid-humid climate (Cong et al. 2013). As

we know, SM and precipitation can interact with each other (Li et al. 2020), which also represents that the variability of China SM in space and time are complex and further takes serious challenges for estimating China SM data based on *in-situ* SM stations.

Previous studies have already produced many SM gridded products covering China or the world, but mainly based on remote sensing data and only for the surface layer (e.g., Chen et al., 2021, Meng et al., 2021, Song et al., 2022, Wang et al., 2021

and Zhang et al., 2021). However, the daily SM data with high quality (high-resolution seamless spatial distribution, long time periods, and low errors) at multiple layers based on *in-situ* measurements do not exist for China yet. Although Sungmin et al. (Sungmin and Orth 2020) generated the global SM data by ML model which includes China region, only less than 20 *in-situ* SM stations in China were applied, which was hardly ensure the quality of China SM product. In addition, this product's resolution is 0.25 degree, which limits its use in applications requiring high resolution SM.

To fill this research gap, in this study, we aimed to generate high quality gridded SM data in China with *in-situ* measurements based on RF model (Fig.1). The covariates were consisted of static data and time series variables, including ERA5-Land (the land component of the fifth generation of European Reanalysis, Balsamo et al. 2015), USGS (United States Geological Survey) land cover type (Loveland et al. 2000), USGS DEM (Digital Elevation Model, Balenović et al. 2016), reprocessed MODIS LAI (Moderate-resolution Imaging Spectroradiometer Leaf Area Index, Yuan et al. 2011) and CSDL

(China Soil Dataset for Land surface modeling, Shangguan et al. 2013). The *in-situ* SM observations from 1,789 stations after quality control procedures were acted as our target variables, which were obtained from China Meteorological Administration (CMA).

Our new China gridded SM product (named SMCI1.0, Soil Moisture of China by *in-situ* data, version 1.0) provides SM data at ten layers, which include soil depth from 10cm to 100cm with an interval of 10cm. Meanwhile, SMCI1.0 has ~1km (30

seconds) spatial resolution and daily temporal resolution over the period from 1 January 2010 to 31 December 2020. For the SMCI1.0 product, we mainly considered the four research questions as follows:

(1) How are *in-situ* SM and all the covariates related, including meteorological data (air temperature, precipitation, total evaporation, potential evaporation), soil data (SM and soil temperature at different soil layers, and static soil properties), leaf area index and land cover type.

(2) Can the RF model successful generate high quality gridded SM (high-resolution seamless spatial distribution, long time periods, and low errors) at multiple layers in China based on *in-situ* SM observations?

(3) How the RF model performs for the space and time extrapolation experiment, in other words, can the RF model generate the SM data with low errors which take *in-situ* SM observations as the reference under year-to-year and station-to-station estimating?

(4) What conditions can SMCI1.0 SM data have lower errors or higher errors against adjusted *in-situ* SM observations?

For the above issues, we make four contributions for generating and validating multi-layer gridded SM data over China. First, we record and make detailed analysis of the correlations between *in-situ* SM and all covariates. Then, we apply the RF to model the complex relationship between covariates and *in-situ* SM observations, and further validate the year-to-year and station-to-station estimating. Finally, we intuitively display and analysis the quality of SMCI1.0 with different conditions,

and it is expected to help researchers improve the China gridded SM intentionally and strategically.



The schema of this work is listed below. Section 2 describes the *in-situ* SM, data served as covariates, RF model and its application in SM estimating. Section 3 gives the validation results, experimental results, a sampled map on a day and relative importance of covariates. Section 4 and 6 present the discussion conclusions, respectively.

## 2.Materials and Methods

### 2.1 *in-situ* SM observations

Target SM data for RF model was constructed from the CMA SM observations. The observations contain 1,789 stations over China (18-N, 73-W) and have hourly temporal resolution over the period from 1 January 2010 to 31 December 2020. The spatial distribution of observations is shown in Fig. 1(a). For our *in-situ* SM observations, two aspects deserve to be noted: one aspect is the large number of in situ stations (i.e., 1,789), which can help ML models to capture the complex nonlinear relationship between SM and covariates over various training conditions and thus to generate high quality China gridded SM data. The other aspect is the bias and standard deviation correction of *in-situ* SM, which is vital for our study to allow the ML model to achieve the high-quality SM product. We applied the same correcting method with that of Sungmin et al. (2020), who adjusted the raw *in-situ* SM observations to match means and standard deviation of the ERA5-Land gridded SM data at the corresponding time periods, grid cells and layers.

The automated quality control of *in-situ* SM observations was performed before training the RF model. We first removed the null values over the long period (10 days timestep) and unreasonable SM values. In checking the unreasonable SM values, four plausibility checks were performed, such as checking geophysical consistency using precipitation and soil temperature, spike detection, break detection and constant values detection. The details could be found in the Global Automated Quality Control method (Dorigo et al. 2013). Finally, the removed values were replaced by the linear method according to the remaining SM values at the same time period from five days ahead and five days later. To facilitate generating 1km gridded SM data at multiple layers by the RF model, the CMA SM observations were processed to daily and the observations were averaged if there are more than one stations within a grid at 1km resolution. We simply average all the available observations in each day at each *in-situ* SM measurement station for daily resolution and all the *in-situ* SM measurement stations if there are more than one stations in each grid for 1km resolution. In this way, we got ~1,789 spatial points (or grids) of observations. The details for the target *in-situ* SM are represented in Fig. 2. Fig. 2(a) shows that stations are dense in the east part of China, but sparse in the west part. Fig. 2(b) represents that the sample size varies with soil depth, and large numbers of missing values exist at 70 and 90 cm soil depths. From Fig. 2 (c), we could see that the values of the *in-situ* SM at all soil depths were mainly concentrated in the range from 0.2 to 0.4 $m^3/m^3$. Fig. 2(d) denotes that the data number in low standard deviation (0~0.05 $m^3/m^3$) is smaller than that in high standard deviation (0.05~0.07 $m^3/m^3$) from at 10 to 40 cm soil depths. But the opposite conclusion can be drawn from 50 to 100 cm soil depths (larger data number in low standard deviation is than that in high standard deviation). Meanwhile, Fig. 2(d) also hints that the standard deviation of SM at deeper soil depth (except that at 100 cm soil depth) is lower than that at upper soil depth. Decreasing standard deviation with increased soil depth denoted that *in-situ* SM is more stable in deep soil depth, which is consistent with the previous studies (Gao and Shao 2012; Wang et al. 2013). From Fig. 2 (e), we could see that the stations have 8 climate types, most observations belong to temperate climate with dry winter (Cw), temperate climate, fully humid (Cf) and snow climate with dry winter (Dw), and the data with tropical monsoon climate (Am) and snow climate, fully humid (Df) are sparse, which occupy only small parts of China.

After generating daily SM based on CMA SM observations for each 1km grid where there is one or more *in-situ* stations, we started to perform the correction of deviation and variance for *in-situ* SM. *in-situ* SM data was obtained by various sensor



types, which had different calibrations. Hence, to overcome the artifacts during the RF model training, we adjusted the observations to match means and standard deviation of the ERA5-Land SM at the corresponding time periods and grid cells (Sungmin and Orth 2020). This method made the target *in-situ* SM resemble the mean and standard deviation of ERA5-Land SM, and kept daily temporal variations which follow the original *in-situ* SM time series. As the soil depth of each soil layer of ERA5-Land SM was inconsistent with that of *in-situ* SM, we mapped the soil layer of ERA5-Land SM to the

corresponding soil layers of *in-situ* SM. Hence, the *in-situ* SM from 10 cm to 30 cm were adjusted based on the gridded SM at layer2 from ERA5-Land dataset (7-28 cm), and the *in-situ* SM from 30 cm to 100 cm were adjusted based on the gridded SM at layer3 from ERA5-Land dataset (28-100 cm).

### 2.2 Datasets as covariates

Table 1 shows the datasets uses covariates used for RF modeling. Most covariates were collected from the ERA5-Land

reanalysis dataset, which was produced by the land component of European Centre for Medium-Range Weather Forecasts (ECMWF). The reasons for selecting the ERA5-Land dataset as preference were as follows: (1) it is generated under a single simulation of a land surface model using ERA5 reanalysis as the forcing data, but with a series of improvements making it more accurate for all types of land applications (Albergel et al. 2018); (2) there are only several months latency for obtaining ERA5-Land datasets, which allowed us to update SMCI1.0 in time; (3) the data is long-term (since 1981) and with seamless

spatial distribution and multilayers, which helps us to generate high quality SMCI1.0 easily. Compared with satellite observations, we can avoid the spatial-temporal gaps and limited time periods covered by using ERA5-Land reanalysis (Sungmin and Orth 2020). The static data of covariates were collected from USGS land cover type (Loveland et al. 2000) and DEM (Balenović et al. 2016), reprocessed MODIS LAI Version 6 for land surface and climate modelling (Yuan et al., 2011) and the China Soil Dataset for Land Surface Modeling (CSDL, Shangguan et al., 2013), including sand, silt and clay

content, rock fragment, and bulk density. The reprocessed MODIS LAI Version 6 was improved by a two-step integrated method that had the advantage of continuity and consistency in space and time series (Yuan et al., 2011). It was worth noting that the temporal resolution of reprocessed MODIS LAI Version 6 was 8 days, and the daily LAI between the 8 days was computed by linear interpolation of the nearest two LAI at 8-day timestep. CSDL was developed for use in the land surface modeling. The spatial distribution of soil type, rock fragment, and bulk density was derived by the polygon linkage method,

which were well represented and consistent with common knowledge of Chinese soil scientists (Shangguan et al., 2013).

### 2.3 Random Forest regression

Random Forest (RF) is an ensemble machine learning approach, which apply the decision trees and bagging methods for the classification and regression problem (Breiman 2001). The simple decision trees model partitioned the variable space and further grouped dataset recursively based on similar instances. For the candidate variables from a set of covariates, a split

was determined by the values of interesting variable that evolved into a tree structure with multiple parent and child nodes. Meanwhile, the response variance for decision regression trees was applied as the criterion for maximizes the purity of each node (the response variance was applied to measure node purity) and further find the optimal split. RF generated diverse decision trees to avoid overfitting through bagging method, which constructed multiple training sub-dataset by resampling with replacement of the original dataset. For each training sub-dataset, a decision tree was growing until the selected

criterion was reached (the value for the minimum node size). After all the decision trees were generated, the average was taken from all the estimations from each decision tree.

The importance of the covariates obtained by the RF model was also worth noting, which computed by a permutation scheme. In the permutation method, the different SM was estimated by permuting all the covariates. Hence, the importance



of covariates could be obtained by comparing their accuracy of SM estimation. Such as, if one covariate was vital to estimate

target SM, the accuracy was expected to decrease for estimation by the remaining non-permuted covariates without the covariate.

### 2.4 The application for Random Forest model

In our study, we first selected the optimal values of hyper-parameters in RF model based on the 10-fold cross-validation method. After selecting the optimal hyper-parameters, two independent experiments are conducted to investigate the

estimation accuracy of the SMCI1.0 at spatial-temporal scale (year-to-year and station to station experiments). In the year-to-year estimating, the data from 2010 to 2017 years in each station was reserved for training set, and to evaluate the estimation accuracy of SMCI1.0 at temporal scale, we compared the generated SM by RF model at each soil depth with the corresponding *in-situ* SM from 2018 to 2020 years. In the station-to-station estimating, the data from 2/3 of the stations with randomly selection from 2010 to 2020 was applied for training the RF model, and the remained 1/3 of the stations were used

to evaluate the estimation accuracy of SMCI1.0 at spatial scale. Finally, the SMCI1.0 product was generated by RF model at 1km, which was built based on the *in-situ* SM and the combined covariates (shown in Table 1) from all stations and all years. In addition to the 1 km resolution, we also produced a version of 9 km resolution by aggregating the higher resolution covariates for the convenience of applications which need only coarser SM. SMCI1.0 can be accessed at http://dx.doi.org/10.11888/Terre.tpdc.272415.

The number of random selected candidate variables from all the covariates (*max_features*) and the value for the minimum node size (*min_samples_leaf*) in RF model were the vital hyper-parameters which affect the performance. The values of *max_features* and *min_samples_leaf* directly determined how the RF model grown. Other hyper-parameters, such as number of trees (*n_estimators*), were not tuned but simply determined based on RF's own training. The hyper-parameters *max_features* affected the split SM values and *min_samples_leaf* was acted as the criterion for stopping the decision tree

growing. Meanwhile, we applied the 10-fold cross-validation method to tune the values of *max_features* and *min_samples_leaf*, and they were selected from range [1,25] with a single interval and [5,30] with 5 intervals via grid hyper-parameters method for preventing RF model over-fitting, which randomly divided the whole dataset into *k*-fold and a 10th of the sub-datasets was used as validation sample while the other sub-datasets were applied for training RF model. The root means square error (*RMSE*) was assessed for evaluating model accuracy by the 10-fold cross-validation method. The

accuracy of RF models with all hyper-parameters based on grid hyper-parameters method at 10 cm soil depth were shown in Table 2. We could see that the RMSE obtained based on all the hyper-parameters ranged from 0.601 to 0.637 and the best accuracy (*RMSE*=0.601) can be achieved when *max_features* and *min_samples_leaf* set to be 1 and 20, respectively. The optimal hyper-parameters (*max_features*=1 and *min_samples_leaf*=20) in RF model were used for further research.

The quality of SMCI1.0 product was evaluated in terms of *ubRMSE*, *MAE* (Mean Absolut Error), *R* (correlation coefficient),

$R^2$ (explained variation) and *Bias*, respectively. *ubRMSE* and *MAE* were applied to test the ability to estimate volatility and fluctuation amplitude, respectively. *R* denotes fluctuation pattern and $R^2$ represents the percentage of variance explained by the RF model *Bias* was used to observe if the estimations were overestimated or underestimated. The five metrics were computed as follows:

$$ubRMSE = \sqrt{\frac{\sum_{i=1}^{N}[(x_i - \bar{X}) - (y_i - \bar{Y})]^2}{N}}, \tag{1}$$

$$MAE = \frac{\sum_{i=1}^{N}|x_i - y_i|}{N}, \tag{2}$$

$$Bias = x_i - y_i, \tag{3}$$



$$R = \frac{\sum_{i=1}^{N}(x_i - \bar{X})(y_i - \bar{Y})}{\sqrt{\sum_{i=1}^{n}(x_i - \bar{X})^2}\sqrt{\sum_{i=1}^{n}(y_i - \bar{Y})^2}}, \tag{4}$$

$$R^2 = 1 - \frac{\sum_{i=1}^{N}(y_i - x_i)^2}{N\sum_{i=1}^{N}(y_i - \bar{Y})^2}, \tag{5}$$

where $y_i$ and $x_i$ denoted the $i$-th *in-situ* SM and gridded SM for all the stations and periods, respectively. $\bar{Y}$ and $\bar{X}$
represented the mean values of the *in-situ* SM and gridded SM, respectively.

## 3.Results

### 3.1 Random Forest model validation

To validate the performance of RF model for generating SMCI1.0, we mainly discussed the modeling ability by year-to-year and station-to-station experiments, which could ensure that SMCI1.0 product has low errors in both temporal and spatial
scale against *in-situ* SM. Meanwhile, we also compared with the state-of-the-art global gridded datasets such as ERA5-Land, SMAP-L4 and SoMo.ml datasets.

The scatter plot between the mean of SMCI1.0 and that of *in-situ* SM at each station, the frequency distributions of all SM values in SMCI1.0 and that in *in-situ* measurements, and the violin-plot for the distribution of daily SM from stations for each climate type were represented in Fig. 3 (from 10 to 30 cm soil depths) and Fig. S1 (from 40 to 100 cm soil depths). As
shown in Fig. 3 (a), we can conclude that there was generally a good agreement between the mean of SMCI1.0 and that of *in-situ* SM at each station (the correlation ranges from 0.867 to 0.908), which demonstrated that the RF model can well capture spatial variations in *in-situ* SM. The RF model showed somewhat better results in deeper soil depths, such as the RF model at 30 cm soil depth had better performance than that at 10 and 20 cm soil depths in Fig. 3 (a), which was consistent with the previous studies (Sungmin and Orth 2020). And the different result was achieved by the RF model at 70 cm and 90
cm soil depths in Fig. S1 (a), where the performance was the worst in all the soil depths (*ubRMSE*=0.053, *MAE*=0.038, *R*=0.867, *R²*=0.731 at 70 cm soil depth; *ubRMSE*=0.052, *MAE*=0.036, *R*=0.883, *R²*=0.759 at 90 cm soil depth). Meanwhile the best result was achieved by the RF model at 30 cm soil depth (*ubRMSE*=0.043, *MAE*=0.033, *R*=0.908, *R²*=0.824 at 30 cm soil depth). The reason may be that RF model is difficult to estimate accurate SM for only a few *in-situ* SM stations. From Fig. 1 (b), we can see that the total numbers of data at 70 cm and 90 cm soil depths is relatively small. In other words, more
diversity of data was expected to help RF model 'learn' complete relationship between covariates and *in-situ* SM and further generated SMCI1.0 with low errors in China. Meanwhile, it also showed the superior quality for our SMCI1.0 product, because the larger numbers of *in-situ* SM data in China were applied for estimating seamless SM than that by the previous studies (Sungmin et al. 2020). In Fig. 3 (b), although the SMCI1.0 had smaller variability in the values range from 0 to 0.18, 0.38 to 0.43, and 0.46 to 0.6 and larger variability in other value ranges, as a whole, SM in SMCI1.0 generally agreed well
with *in-situ* SM. The same conclusion can be drawn from 40 to 100 cm soil depths in Fig. S1 (b). The SMCI1.0 were further evaluated for each climate type in Fig. 3 (c) and Fig. S1 (c). With regard to the violin-plot, RF model can estimate consistent results with *in-situ* SM. However, the inconsistent SM was estimated in Tropical Monsoon Climate (Am) and Desert Climate (Bw). The reason could also be attributed to only few *in-situ* SM in these climatic regions, which represented in Fig 1 (e). Finally, we concluded that RF model can reproduce the temporal variation in *in-situ* SM at unseen period accurately.
Meanwhile, we also advocated that more diverse training data over various regions was needed for capturing the complex relationship between covariates and SM, and further improving the quality of high resolutions SM product.

From Fig. 4 and Fig. S2, we could see that although the results of the station-to-station experiment were inferior to that of the year-to-year estimating, RF model can also perform well in estimating seamless SM in China at unseen locations.





Additionally, similar to the year-to-year experiment, RF model performed the best at 30 cm soil depth than that at other soil
depths in the station-to-station experiment.

Finally, we also compared SMCI1.0 product with other gridded datasets (ERA5-Land, SoMo.ml and SMAP-L4) according
to the median *ubRMSE*, *R*, *Bias* and *MAE*. From Fig. 5 and Fig. S3, SMCI1.0 product had the lowest median *ubRMSE* and
*MAE* from 10 cm to 100cm soil depths. Regarding the median *Bias* between gridded SM and *in-situ* SM observations,
SMCI1.0 product had almost similar quality with ERA5-Land datasets for all the soil depths, but had higher quality than
SoMo.ml and SMAP-L4 datasets. It was worth noting that the SMAP-L4 dataset had the widest spread of errors and tended
to underestimate *in-situ* measurements, which leaded to higher median *ubRMSE* and *MAE* values. Regarding the median *R*
between gridded SM and *in-situ* SM observations, SMCI1.0 product had slightly higher quality than SoMo.ml dataset for
10cm, 20cm, 80cm and 100cm soil depths and obvious advantages than ERA5-Land and SMAP-L4 datasets for all the soil
depths, while it had lower quality than SoMo.ml dataset for other soil depths. Considering all the above metrics, SMCI1.0
product were more robust than the other gridded datasets. Interestingly, it was inconsistent for the results of *R*, *ubRMSE*, and
*MAE* in Fig. 3 and Fig. 5, which had the same phenomenon with the previous studies (Sungmin and Orth 2020) (represented
in their Fig. 5 and Fig. 6). For example, SMCI1.0 product had the *ubRMSE*, *MAE* and *R* being 0.046, 0.035 and 0.889 at 10
cm soil depth in Fig. 3. However, in Fig. 5, the box-plot represented the lowest *ubRMSE*, *MAE* and highest *R* of SMCI1.0
product were nearly 0.03, 0.02, and 0.7, respectively. The reason may be that the same metrics were calculated in different
ways, the one in Fig. 3 was to count the results of all stations and temporal period, and the one in Fig. 5 was to count the
results of only temporal period at one station.

Overall, the RF model can successfully generate the SM data with low errors taking *in-situ* SM observations as the reference
at unseen periods and locations. SMCI1.0 product outperforms the existing SM products (ERA5-Land, SoMo.ml and SMAP-
L4) in the sense of statistic metrics.

### 3.2 The spatial and temporal evaluation of the SMCI1.0

As the section 3.1 evaluated the overall performance of estimated SM at the macro level, the variability and trends of the
SMCI1.0 in temporal and spatial scale cannot be reflected. Hence, to take the evaluation of the SMCI1.0 in temporal scale,
we randomly selected stations from different climate regional for evaluating the SM temporal dynamics of the SMCI1.0,
ERA5-Land, SMAP-L4, SoMo.ml and *in-situ* SM from 10 cm to 20 cm soil depths. And for the spatial scale, we represented
the estimation performance for each *in-situ* SM station in terms of *ubRMSE*, *R*, and *bias*, respectively. Noticeably, in order to
evaluate each station as much as possible, we apply year-to-year experiment in this testing.

Fig. 6 compared the SM temporal dynamics of the SMCI1.0, ERA5-Land, SMAP-L4, SoMo.ml, and *in-situ* SM at 10 cm
soil depth along with local precipitation. We could see that although the SMCI1.0 product had large deviation compared with
*in-situ* SM in snow climate, fully humid zone (Df-51431:E, N), it was almost consistent with *in-situ* SM in other regions. It
was necessary to note that the SM in Desert Climate region (Bw-W1063:E, N) had high variability but with low precipitation
from 231th to 325h days, the SMCI1.0 product could still adequately capture their relationship (represented in the light blue
rectangle). Overall, the SMCI1.0 could follow the reasonable patterns which *in-situ* SM increased with wet condition and
decreased with dry conditions. During the rainfall near 91[th] day across the Tropical Monsoon Climate zone (Am) and near 1[st]
day across the Snow climate with dry winter zone (Dw), the *in-situ* SM did not increase with high precipitation, but the
SMCI1.0 product could capture the increase in SM (denoted in the light blue rectangle). The reason may be that the applied
covariates had bias with *in-situ* measurement and further affected estimation by RF model. Meanwhile, we also found the RF
model could overcome much bias in dry conditions, except for that from 196[th] to 305[th] days in the snow climate, fully humid
zone (shown in the light red rectangle). In the case of 30 cm soil depth (represented in Fig. S4), we could see an agreement





between several peak events, it could be attributed to the soil texture homogeneity in the 10 and 30 cm soil depths. Almost
all climatic regions had lower dynamic ranges at 30 cm soil depth than that at 10 cm, this may be attributed to the persistent
behavior of SM at 30 cm soil depth. For the evaluations of SM temporal dynamics from 10 to 30 cm, we can see that
SMCI1.0 can broadly capture the temporal characteristic of *in-situ* SM and further demonstrated the high quality of SMCI1.0
product.

Fig. 7 represented the *in-situ* testing performance according to the fit statistics (*ubRMSE*, *R*, *Bias*, and *MAE*). We could see
that the SMCI1.0 product had relatively low *ubRMSE*, *Bias*, and *MAE* over most regions. In combination with Fig. 8, we also
found that the low errors of SMCI1.0 product were often in the arid regions, which was consistent with the previous study
(Zhang et al. 2019). However, the higher *ubRMSE*, *MAE* and lower *R* could be seen in North China Monsoon Region. The
North China Monsoon Region has typical temperate monsoon climate characteristics, where the annual temperature is high
and the rainy season is concentrated. The SM variations in the North China Monsoon Region were complex, which may
present great challenges for estimating SM by RF model. Despite SMCI1.0 product had lower *R* in North China Monsoon
Region than that in other climatic regions, the *R* values were mostly larger than 0.5 (within the acceptable limit). This
highlighted the robustness of SMCI1.0 product. According to the *Bias* in Fig. 7, we could see that SMCI1.0 product tends to
be underestimated in the northeast and southwest China, and be overestimated in the east China, which had the similar trend
with ERA5-Land dataset and we could also draw the similar conclusions for the box-plot of Bias in Fig. 5. Meanwhile, it had
the opposite estimations with SoMo.ml dataset in north China and Sichuan province (SMCI1.0 product often underestimated
in north China and overestimated in Sichuan province, but SoMo.ml dataset was the opposite), but SMCI1.0 product had
lower errors in estimating *in-situ* SM. According to the *R* in Fig. 7, SMCI1.0 product had the similar results with SoMo.ml
dataset, and performed better than ERA5-Land and SMAP-L4 datasets, which could also be represented by the box-plot of *R*
in Fig. 5. In the case of 30 cm soil depth in Fig. S5, the SMCI1.0 product had higher accuracy than that at 10 cm soil depth,
especially in terms of *ubRMSE* and *MAE* metrics. The reason may be the background aridity leaded to low variability of SM
in the deeper layers (Karthikeyan and Mishra 2021). The RF model can capture the variation in SM easier.

**3.3 Spatial patterns of SMCI1.0**

To describe the general spatial patterns of SMCI1.0 over the China, we presented the SM maps at 1km spatial resolution for
1st January 2016. From Fig. 8, we could see that the spatial contiguity of SM patterns for SMCI1.0 was captured well, and
most high-resolution details of SM patterns in all the climatic region for SMCI1.0 had more detailed "expression" than that
for other SM products. Meanwhile, the spatial pattern of SMCI1.0 is consistent with those of high-resolution covariates such
as DEM and LAI in some regions, which also denoted that the SMCI1.0 could better reflect the detailed spatial distribution
of SM. Southeast China is the tropical monsoon climate zone, where the rainy season was concentrated (represented in Fig.
6). Hence, these regions are predominantly wet in the SM maps. Northwest China is the Desert Climate region, which had
not any rainfall and further lead to the dry conditions (also represented in Fig. 6). Qinghai province belongs to the tundra
climate zone, where some soils are wet and other soils are dry. This is probably due to the complicated topography of
Qinghai Province that some regions with woody plants can intercept rainfall, which may decrease the overall water input
into the soil (Zwieback et al. 2019), and other regions with vegetation can decreases soil temperature and evaporation from
the soil surface by shading, which avoid the loss of soil moisture (Kemppinen et al. 2021).

**3.4 Relative importance of covariates**

The relative importance of covariates at the ten soil depths was shown in Fig. 9 and Fig. S6. Bars represented the variability
of relative importance across the covariates. As represented in Fig. 9, the ERA5-Land SM was the most important to

estimate *in-situ* SM from 10 to 100 cm soil depths. In addition to ERA5-Land SM covariates, evapotranspiration, DEM, clay, reprocessed MODIS LAI (Version 6), porosity, LAI low vegetation, air temperature, LAI high vegetation and silt were

followed. The importance of other covariates was less than 0.01, which were not detailed discussed in this study. As we know, had strong correlation with SM dynamic under water-limited conditions (Albertsona and Kiely 2001). So, evapotranspiration had greatly associated with SM in the regression model. Clay, porosity, rock fragment, silt and sand were properties in the soil. Bissonnais et al. (Bissonnais et al. 1995) tested SM for 31 soil types with different soil properties over Illinois region and denoted that the available SM varied by each soil group. They could help RF model identify variation in

SM through different soil properties. LAI was a vital parameter in the land surface and controlled many complex processes in relation to vegetation, which determined evapotranspiration and further had impact on water balance (Chen et al. 2015). It is worth note that reprocessed MODIS LAI (Version 6) (Yuan et al. 2015) had larger impact on SM estimation than the LAI of reanalysis products. The reason may be that it had better quality than the LAI of reanalysis products. Air temperature and SM were closely related, such as the climate shifts from the hot to the cold, SM decreased for all land covers (Feng and Liu

2015). However, air temperature had significant effect in the RF model for upper soil layers (at 10 cm and 20 cm soil depths) while it began to weaken in the deeper soil (represented in Fig. S6), which was consistent with the previous studies (Hu and Zheng 2003). Interestingly, as widely known, the land cover type is highly related to the variation in SM. However, it had relative low importance (less than 0.01) for the RF model than the above covariates. Noticeably, its importance was computed at the 1 km spatial resolution, the different importance of land cover type may be found at higher spatial resolution.

Such as land cover type had less important to SM at coarse spatial resolution (Gaur and Mohanty 2016; Joshi et al. 2010), but had strong correlation with *in-situ* SM (Baroni et al. 2013). Meanwhile, intuitively, precipitation was also closely related, SM-precipitation coupling had received increasing interest in recent years (Seneviratne et al. 2010). Although the importance of precipitation (less than 0.01) was not reflected in the RF model, this did not imply that precipitation had not impact on the variation in SM. This could be attributed to the relatively small frequency for daily rainfall during several years periods,

which led to a low ranking compared with other covariates based on the selection metric of RF importance ranking. It should be noted that the static variables and the reprocessed LAI provide information at 1km or 500m resolution, while ERA5-Land is at 9km resolution. So, the spatial details under 1km resolution came from the static variables and the reprocessed LAI rather than ERA5-Land. This aspect cannot reflect by the importance of RF as RF models were established to mainly reflect the temporal variation. This is because that we have much more samples of SM in the time dimension than those in the

spatial dimension (1,789, the total number of stations). As a result, the importance of higher resolution variables (especially static variables) in estimating the spatial variation of SM was essentially underestimated by the importance of RF.

## 4.Discussion

### 4.1 The quality of SMCI1.0 at spatial-temporal scale

In this study, the gridded soil moisture was estimated through RF method in China based on the ERA5-Land reanalysis,

USGS land cover type and DEM, reprocessed LAI and soil properties from CSDL, which included soil depths from 10cm to 100cm and had 1km spatial and daily temporal resolution over the period from 1 January 2010 to 31 December 2020. The training efficiency was high (*RMSE*=0.601) due to the selection of important factors and vital hyper-parameters (*max_features*=1 and *min_samples_leaf*=20). In the year-to-year experiment, the *RMSE*, *MAE*, *R* and $R^2$ between gridded soil moisture and *in-situ* soil moisture ranged from 0.041-0.052, 0.03-0.036, 0.883-0.919 and 0.767-0.842, respectively. In

the station-to-station experiment, the *RMSE*, *MAE*, *R* and $R^2$ between gridded soil moisture and *in-situ* soil moisture ranged from 0.045-0.051, 0.035-0.038, 0.866-0.893 and 0.749-0.798, respectively.



### 4.2 Requirement of further validations

SMCI1.0 product generally agrees with *in-situ* SM in China than other datasets in general, under the validations with year-to-year and station-to-station. However, we cannot ensure the same quality of SMCI1.0 product in the whole China. The
reason is that *in-situ* SM stations are uneven distribution, and the *in-situ* SM in the western China is sparse. We hope more *in-situ* SM stations are evenly deployed in China, which can ensure the quality of SM in most regions as far as possible. Triple collocation analysis (Karthikeyan and Mishra 2021) is also an alternative method for evaluating SMCI1.0 product. Meanwhile, there are many possible reasons for the failure of RF model, such as insufficient data and the 'learning ability' of model-self. Hence, not only additional records from China are needed to be available, but also more robust estimated models
are hoped to explored. Such as, the single deep learning model are built and optimized in each homogeneous region (Karthikeyan and Mishra 2021), or the optical remote sensing should be used for the human-induced regions (Chen et al. 2021), which can better estimate SM.

### 4.3 Higher-resolution SM estimating

As we know, higher-resolution SM estimation is typically considered as a challenging task (Peng et al. 2020). The relative
important covariates can help estimating model enhance the quality of higher-resolution SM product. The SMCI1.0 product may be acted as a vital covariate for improving the higher-resolution (<1km) SM product. Next, high-resolution SM product generated based on the lower-resolution covariates can also understand as super-resolution task in the computer science, the advanced deep learning models with high performance can also be explored (Lei et al. 2020; Zhang et al. 2020; Zhu et al. 2021). Of course, the target *in-situ* SM with dense distribution is also needed, thus can ensure the quality of high-resolution
SM and further provide the reliable validation.

### 4.4 Sensitivity to precipitation and air temperature

We applied partial correlation to analysis the sensitivity between the meteorological variables (precipitation, air temperature and radiation) and SM. As Fig. 10 shown, precipitation had stronger correlation with SM in SMCI1.0 and ERA5-Land than that in SoMo.ml product across most regions in China, and it represented significant positive partial correlations.
Additionally, air temperature had significant positive partial correlations with SM in the northwestern China, and negative partial correlations in north China and Liaoning province for SMCI1.0. The results with negative partial correlations between air temperature and SM were consistent with the physical knowledge that higher evaporation may be caused by higher air temperatures, and they also leaded to lower SM. In some of the plateau areas, the shortwave radiation was the dominant factors of SM variability for SMCI1.0 product, it had the consistent physical knowledge which the strong radiation in the
plateau area had a great impact on the land surface process. Meanwhile, we also found that the shortwave radiation had the great influence on the SM variability in Tropical Monsoon Climate regions, which was also consistent with the previous studies (Yao et al. 2011). The negative correlation between radiation and SM for SoMo.ml product in Temperature Climate region was stronger than that for SMCI1.0 product, which could explain more negative trends in SM in Temperature Climate region for SoMo.ml product. Compared with other SM products, the SMCI1.0 had similar spatial patterns for all the partial
correlations. Overall, the SMCI1.0 product had reasonable quality in reflecting the relationship between SM and its related meteorological variables.



## 5.Data and code availability

All resources of RF model, including training and testing code is publicly available at https://github.com/ljz1228/SMCI1.0_RF data with the resolution of 1 km and 9km can be accessed at
http://dx.doi.org/10.11888/Terre.tpdc.272415 (Shangguan et al. 2022).

## 6.Conclusions

High resolution SM has several potential applications in flood and drought prediction and carbon cycle modelling. SM gridded products covering China or the world are currently based on remote sensing data or based on numerical modeling. However, there is still a lack of SM data with high resolution at multiple layers based on *in-situ* measurements for China.
Through this work, we generated a 1 km resolution long-term gridded SM data in China with *in-situ* measurements based on RF model, which has 10 layers up to 100 cm deep at daily resolution over the period 2010-2020.

Two independent experiments with *in-situ* soil moisture as the benchmark are conducted to investigate the quality of SMCI1.0: year-to-year experiment (*ubRMSE* ranges from 0.041-0.052, *MAE* ranges from 0.03-0.036, *R* ranges from 0.883-0.919, and $R^2$ ranges from 0.767-0.842) and station-to-station experiment (*ubRMSE* ranges from 0.045-0.051, *MAE* ranges
from 0.035-0.038, *R* ranges from 0.866-0.893, and $R^2$ ranges from 0.749-0.798). SMCI1.0 generally has advantages over other gridded soil moisture products, including ERA5-Land, SMAP-L4 and SoMo.ml. Meanwhile, with regard to the fit statistics (*ubRMSE*, *R*, *Bias*, and *MAE*), we could see that the SMCI1.0 product has relatively low *ubRMSE*, *Bias*, and *MAE* over most regions. However, the high errors of soil moisture often located in North China Monsoon Region. Moreover, SMCI1.0 has reasonable spatial pattern and demonstrate more spatial details compared with existing SM products. As a
result, our SMCI1.0 product based on *in-situ* data can be useful complements of existing model-based and satellite-based datasets for various hydrological, meteorological, and ecological analyses and modeling, especially for those applications requiring high resolution SM maps. Furter works may focus on improving the SM map by using advanced deep learning methods and adding more observations, especially for the west part of China.

## 7.Author contributions

WSG conceived the research and secured funding for the research. QLL and WSG performed the analyses. QLL wrote the first draft of the manuscript. GSS and QLL conducted the research. WSG and QLL reviewed and edited the paper before submission. All other authors joined the discussion of the research.

## 8.Competing interests

The authors declare that they have no conflict of interest.

## 9.Acknowledgements

The authors are grateful to all the data contributors who made it possible to complete this research.



**10.Financial support**

The study was partially supported by the National Natural Science Foundation of China, grant number 42105144, 41975122 and U1811464.

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




**Figure 1: Generation process for the SMCI1.0 product with 1km spatial resolution and daily temporal resolution over the period from 1 January 2010 to 31 December 2020 over China.**



**Figure 2: (a) The locations of all stations in China; (b) Total data number per soil depth; (c) Frequency of data length per layer for SM values; (d) Frequency of data length per layer for standard deviation; (e) Total data number per climate zone.**



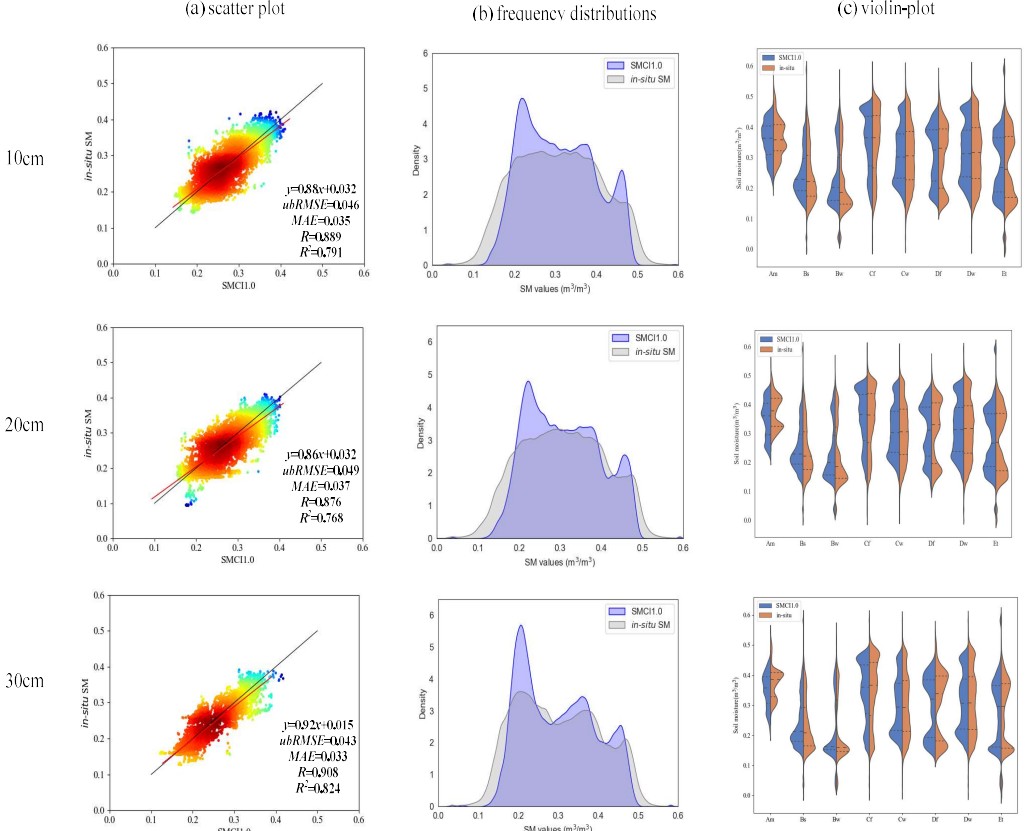

**Figure 3: Comparisons between SMCI1.0 and *in-situ* SM from 10 to 30 cm soil depth: comparison of (a) the scatter plot between the mean of SMCI1.0 and that of *in-situ* SM at each station, (b) the frequency distributions of all SM values in SMCI1.0 and that in *in-situ* measurements, (c) the violin-plot for the distribution of daily SM from stations for each climate type.**



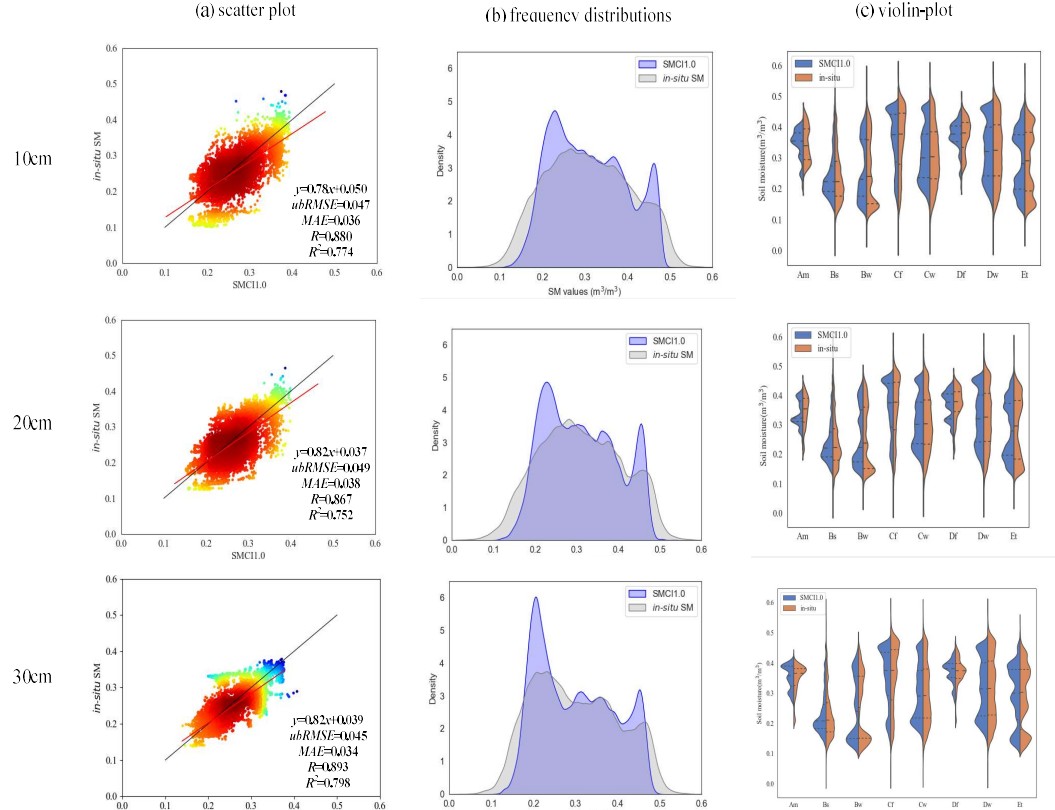

**Figure 4: Same as Fig. 3 but for station-to-station estimating.**

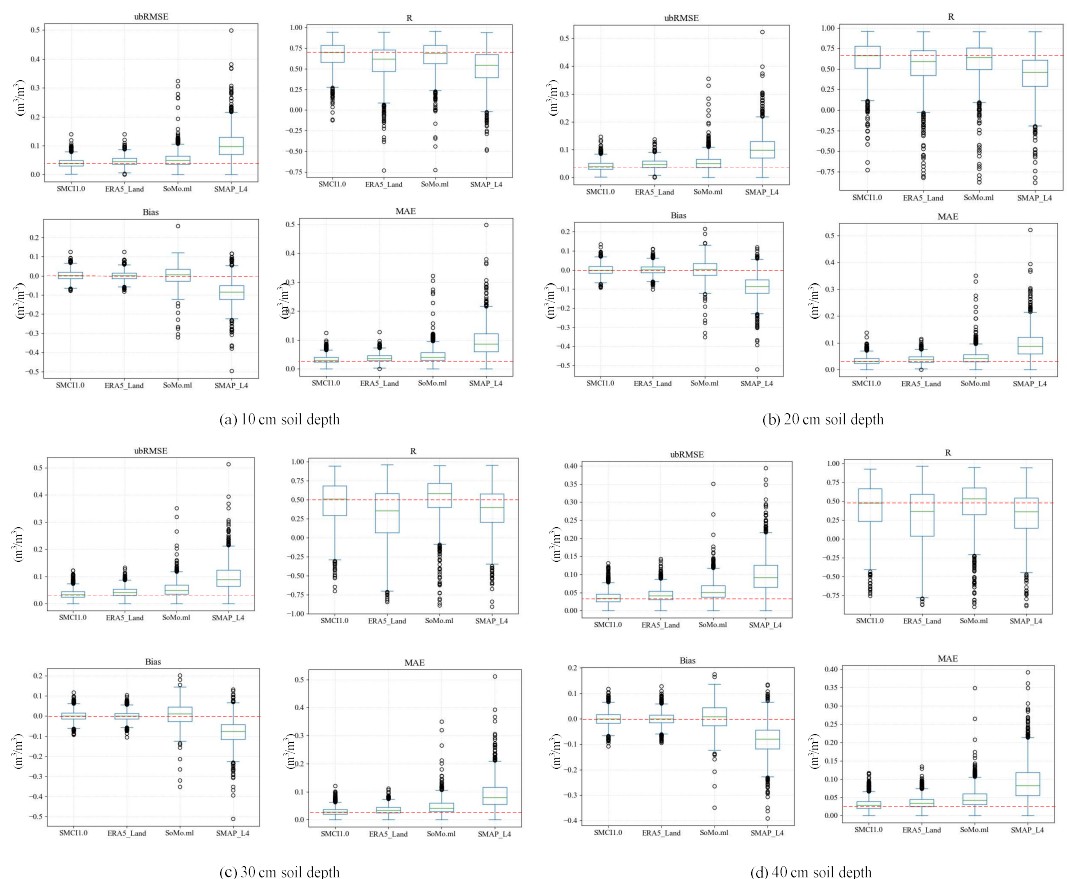


(a) 10 cm soil depth

(b) 20 cm soil depth

(c) 30 cm soil depth

(d) 40 cm soil depth

**Figure 5: Comparison between gridded datasets (SMCI1.0, ERA5-Land, SoMo.ml and SMAP_L4) at soil depths of (a) 10 cm, (b) 20 cm, (c) 30 cm, and (d) 40 cm. The red lines indicate the zero value for Bias and the best performance among datasets for *ubRMSE*, *R* and *MAE*.**






**Figure 6: Time series of *in-situ* and estimated SM by RF model at 10 cm soil depth along with daily precipitation in different climatic zones.**


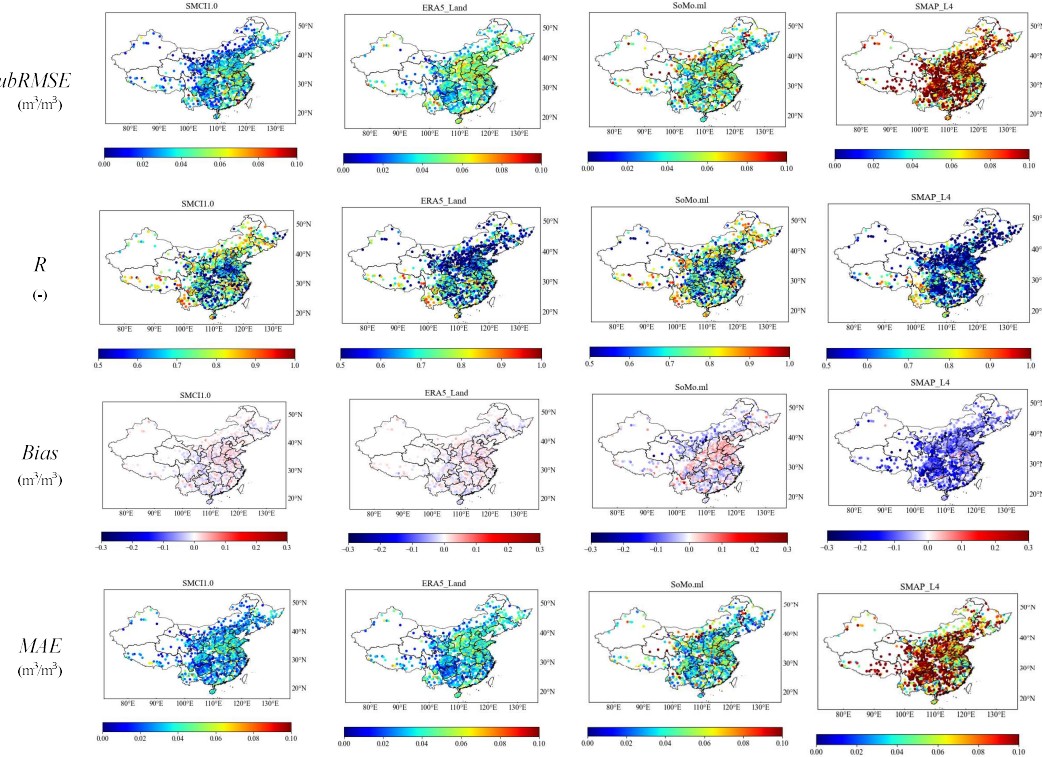

**Figure 7: Goodness of fit statistics (*ubRMSE*, *R*, *Bias*, and *MAE*) at 10 cm soil depth for the RF model during the tested period.**

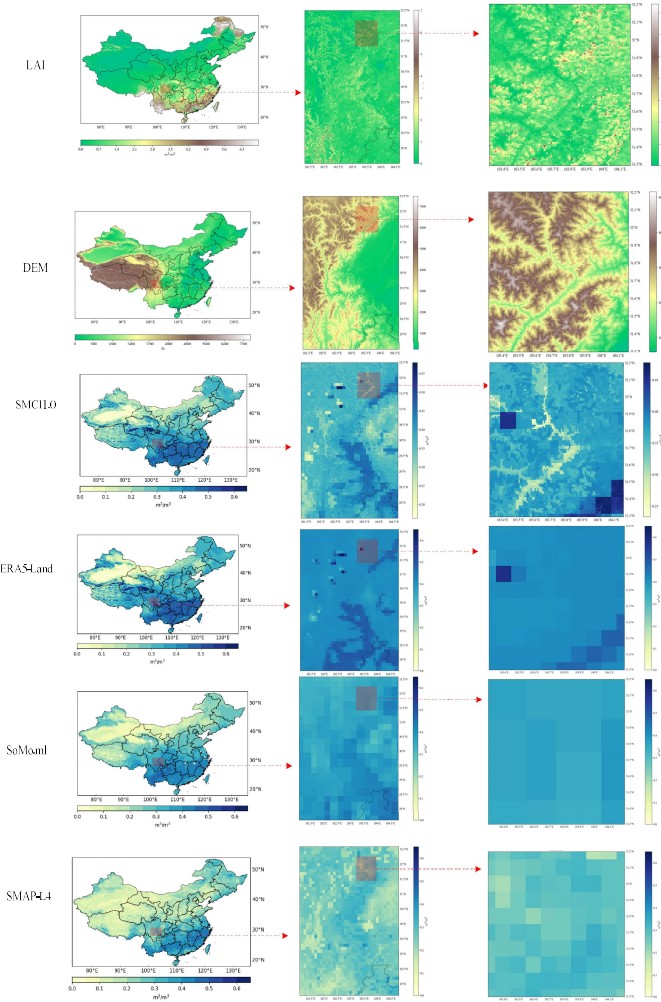

**Figure 8: Soil moisture maps from different products on 1st January 2016. The resolution is 1km for SMC1.0, 9km for ERA5-land and SMAP-L4 and 0.25 degree for SoMo.ml.**






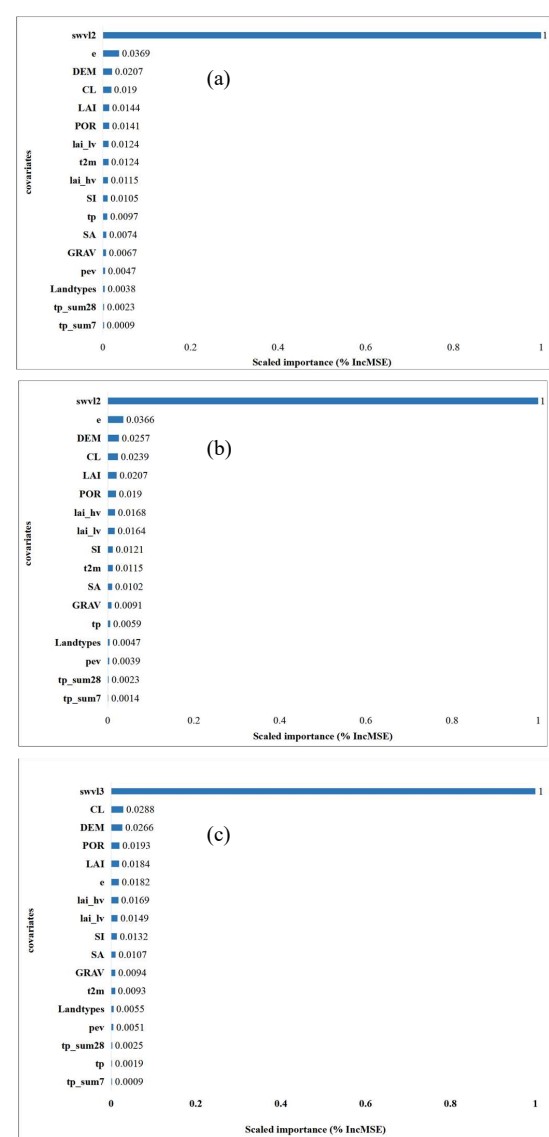

**Figure 9: Relative importance of covariates for the random forest (RF) model at soil depths of (a) 10 cm, (b) 20 cm, (c) 30 cm.**



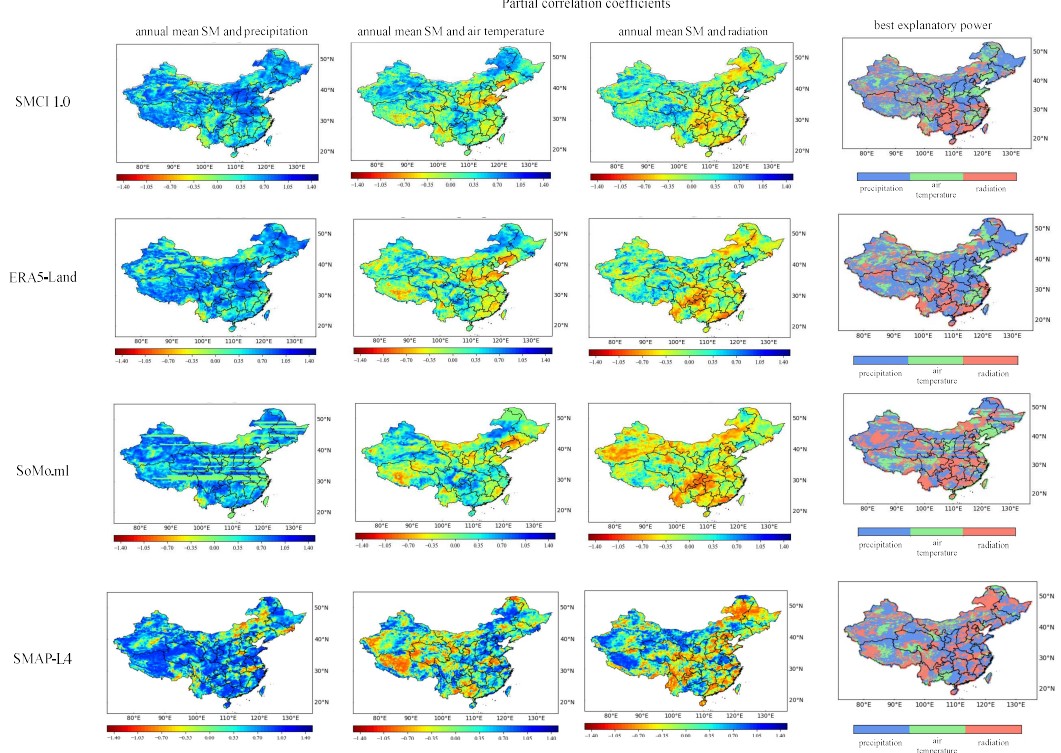


**Figure 10: Partial correlation coefficients between annual mean SM and precipitation (the first column), air temperature (the second column), and radiation (the third column) for the different gridded SM products. The fourth column represents best explanatory power (highest absolute partial correlation) for the interannual variability in SM for the different gridded SM products.**




**Table 1. Details of the covariates for training the Random Forest model.**

| Source | Type | Variable (code) | Description | Time span | Spatial Resolution | Temporal Resolution |
|---|---|---|---|---|---|---|
| ERA5-Land (Land component of the fifth generation of European Reanalysis) | Time series | precipitation (tp) accumulated precipitation in one week (tp_sum7) accumulated precipitation in one month (tp_sum28) air temperature (t2m) potential evaporation (pev) total evaporation (e) leaf area index high vegetation (lai_hv) leaf area index low vegetation (lai_lv) soil moisture from 28 to 100 cm soil depth (swvl2 to swvl3) | meteorological forcings and land surface variables | 2010~2020 | ~9 km | hourly |
| CSDL (China Soil Dataset for Land surface modeling) | Static | rock fragment (GRAV) Porosity (POR) Sand, Silt, Clay (SA, SI, CL) | Soil covariates | --- | ~1 km | --- |
| USGS (Unite States Geology Survey) | Static | Land cover type (Landtypes) Elevation (DEM) | Predominant land cover type and elevation | --- | ~1 km | --- |
| Reprocessed MODIS LAI Version 6 | Time series | Leaf area index (LAI) | Reprocessed LAI using a two-step integrated method | 2010~2020 | ~500 m | 8-day |



**Table 2. The accuracy of the RF models with all hyper-parameters at 10 cm soil depth based on grid hyper-parameters method**
**(the best hyper-parameter is shown in bold font).**

| min_samples_leaf / max_features | 5 | 10 | 15 | 20 | 25 |
|---|---|---|---|---|---|
| 1 | 0.0608 | 0.0603 | 0.0602 | **0.0601** | 0.0602 |
| 2 | 0.0621 | 0.0614 | 0.0611 | 0.0610 | 0.0608 |
| 3 | 0.0625 | 0.0619 | 0.0616 | 0.0614 | 0.0613 |
| 5 | 0.0626 | 0.0622 | 0.0618 | 0.0616 | 0.0615 |
| 6 | 0.0629 | 0.0624 | 0.0621 | 0.0619 | 0.0617 |
| 7 | 0.0629 | 0.0624 | 0.0621 | 0.0619 | 0.0618 |
| 8 | 0.0631 | 0.0626 | 0.0623 | 0.0621 | 0.0620 |
| 9 | 0.0631 | 0.0626 | 0.0623 | 0.0622 | 0.0620 |
| 10 | 0.0631 | 0.0626 | 0.0623 | 0.0622 | 0.0620 |
| 11 | 0.0631 | 0.0627 | 0.0624 | 0.0623 | 0.0621 |
| 12 | 0.0632 | 0.0627 | 0.0625 | 0.0622 | 0.0622 |
| 13 | 0.0632 | 0.0628 | 0.0625 | 0.0623 | 0.0622 |
| 14 | 0.0633 | 0.0628 | 0.0625 | 0.0624 | 0.0622 |
| 15 | 0.0634 | 0.0628 | 0.0626 | 0.0624 | 0.0622 |
| 16 | 0.0634 | 0.0629 | 0.0626 | 0.0624 | 0.0623 |
| 17 | 0.0634 | 0.0629 | 0.0627 | 0.0625 | 0.0624 |
| 18 | 0.0633 | 0.0629 | 0.0627 | 0.0625 | 0.0624 |
| 19 | 0.0634 | 0.0629 | 0.0627 | 0.0626 | 0.0625 |
| 20 | 0.0635 | 0.0630 | 0.0627 | 0.0626 | 0.0625 |
| 21 | 0.0635 | 0.0630 | 0.0628 | 0.0626 | 0.0625 |
| 22 | 0.0635 | 0.0631 | 0.0628 | 0.0626 | 0.0626 |
| 23 | 0.0636 | 0.0631 | 0.0629 | 0.0627 | 0.0625 |
| 24 | 0.0636 | 0.0632 | 0.0629 | 0.0627 | 0.0626 |
| 25 | 0.0637 | 0.0632 | 0.0630 | 0.0628 | 0.0627 |