# Peer review of "A 1-km daily soil moisture dataset over China using in-situ measurement and machine learning"

_Earth System Science Data, 2022_

## Author Comment (AC3)

Note: The modifications are shown in green. The responses to comments are blue colored.

We are very grateful to the reviewer for reviewing the paper so carefully. These comments are very helpful to improve the quality of the manuscript. Please find our itemized responses below and our revisions will be in the revised manuscript.

Comment#1: In this study, only the Random Forest method was applied to derive the upscaled soil moisture data. Is it possible to try more MLs, such as CatBoost, XgBoost, and NeuralNetwork, to test how consistent or different the resulted products are?

Responds:

According to previous studies introduced in the introduction, RF models have proved to be successful in producing SM data and its computation time is acceptable. So, we choose this model in our studies.

According to this comment, we have also tested three more ML models (CatBoost, XgBoost, and Multilayer Perception) and found that the performance of these models is very close to RF. We applied the *scikit-learn* tools to evaluate them. We optimized the model parameters as follows. For the CatBoost model, we used the default parameters in catboost library. For the XgBoost model, we optimize the XgBoost by tuning the parameters through the *selection.GridSearchCV* function which is provided in the *scikit-learn tools*, we set *learning_rate* being 5.0e-3, *max_depth* being 6, *randam_state* being 42, and *estimators* being 100. For the Multilayer Perception, we set *random_state* being 1 and *max_iter* being 500. Take the year-to-year experiment at 10 cm soil depth as an example (see the figure below), RF achieved *ubRMSE* being 0.046, *MAE* being 0.035, *R* being 0.889, $R^2$ being 0.791. The *ubRMSE*, *MAE*, *R*, $R^2$ of CatBoost were 0.045, 0.034, 0.890 and 0.799, respectively. XgBoost achieved *ubRMSE* being 0.045, *MAE* being 0.034, *R* being 0.890, $R^2$ being 0.799. The *ubRMSE*, *MAE*, *R*, $R^2$ of ANN-based model were 0.045, 0.035, 0.891 and 0.796, respectively. According to these results, we don't think it is necessary to use other models instead of RF to produce the high-resolution SM as they cann't outperform it much.

[Figure]

In the modified manuscript, we have added the explanation why only RF model was shown in the discussion. The new expression is as follows:

It was necessary to note that we also compared the RF model with other ML models, including

CatBoost (Dorogush et al. 2018), XgBoost (Chen et al. 2016), and Neural Network (Rosenblatt et al. 1958) based models. We found that the performance of these models is very similar to RF models with a R2 around 0.79. In addition, RF has been widely applied and recognized in SM prediction and many other fields (Carranza et al. 2021, Lin et al. 2022, Ly et al. 2021) and it does not take too much computing time to make the predictions for the whole China. Hence, we only took RF model to produce the high-resolution SM data.

Comment#2: Most of the source datasets cover the period before year 2010. Is there any special reason why the new soil moisture only covers the period 2010-2020? Is it possible to extend the present time period to year 2000-2020?

Responds:

Thanks for your kind comments and helpful suggestions, although most of the applied covariates cover the period before year 2010, we do not access to the *in-situ* measurements before 2010, currently. However, the in-situ measurements before 2010 may be available from China Meteorological Administration (not open to us) and the number of stations is less than 800. If we produce the SM data set without any in-situ data (or only a few hundred stations), the quality of the data may be poorer as it will be extrapolation in time. However, we agree that it is still possible to extend the present time period to year 2000-2020 or even before. So, we list it as a future work in the conclusion as follows:

It is also possible to extent the time coverage of this data set before 2010, even though we do not have access to the in-situ data before 2010 now and the available in-situ stations may be less than 800, which will lead to poorer quality if not enough in-situ data are used.

Comment#3: The "Materials and Methods" read too long, and the authors may try to shorten the text and put some figures into the Supplementary Material.

Responds:

According to this comment, we have put the Figure 2 and related text into the Supplementary Material. We also shortened the text in this section as follows.

In section 2.1, we combined the following text with the last paragraph of this section:
"The other aspect is the bias and standard deviation correction of in-situ SM, which is vital for our study to allow the ML model to achieve the high-quality SM product. We applied the same correcting method with that of Sungmin et al. (2020), who adjusted the raw in-situ SM observations to match means and standard deviation of the ERA5-Land gridded SM data at the corresponding time periods, grid cells and layers." (deleted)
The last paragraph of this section:
After the above data processing, we started to perform the correction of deviation and variance for in-situ SM, which is vital for our study to allow the ML model to achieve the high-quality SM product. In-situ SM data was obtained by various sensor types, which had different calibrations. Hence, to overcome the artifacts during the RF model training, we adjusted the observations to match means and standard deviation of the ERA5-Land SM at the corresponding time periods, grid cells and layers using the same method with that of Sungmin and Orth (2020). This method made

the target in-situ SM resemble the mean and standard deviation of ERA5-Land SM, and kept daily temporal variations which follow the original in-situ SM time series. As the soil depth of each soil layer of ERA5-Land SM was inconsistent with that of in-situ SM, we mapped the soil layer of ERA5-Land SM to the corresponding soil layers of in-situ SM. Hence, the in-situ SM from 10 cm to 30 cm were adjusted based on the gridded SM at layer2 from ERA5-Land dataset (7-28 cm), and the in-situ SM from 30 cm to 100 cm were adjusted based on the gridded SM at layer3 from ERA5-Land dataset (28-100 cm).

We have alsoput the Table 2 into the Supplementary Material and shortened the related paragraph as follows:

The number of random selected candidate variables from all the covariates (max_features) and the value for the minimum node size (min_samples_leaf) in RF model are the vital hyper-parameters which affect the performance. Other hyper-parameters, such as number of trees (n_estimators), were not tuned but simply determined based on RF's own training. Meanwhile, we applied the 10-fold cross-validation method to tune the values of max_features and min_samples_leaf, and they were selected from range [1,25] with a single interval and [5,30] with 5 intervals via grid hyper-parameters method for preventing RF model over-fitting. The accuracy of RF models with all hyper-parameters based on grid hyper-parameters method at 10 cm soil depth were shown in Table 1S. We could see that the root means square error (RMSE) obtained based on all the hyper-parameters ranged from 0.601 to 0.637 and the best accuracy (RMSE=0.601) can be achieved when max_features and min_samples_leaf set to be 1 and 20, respectively, which were used for further research.

We also revised shortened other contents. This will be shown in the revised manuscript.

Comment#4: For the "Results", they seem to be a combination of results analysis and short discussion. Please move relevant discussion content to the "Discussion" part.

Responds:

Thanks for your kind comments and helpful suggestions, we have moved Section 3.4 in the old manuscript to the discussion as Section 4.1, and put some short discussions in the results of the old manuscript into the "Discussion" part. The new section in the discussion is as follows:

Figure 2 and 2s shows that the result at 70 cm and 90 cm were significant worse than those at other depths. The reason may be that RF model is difficult to estimate accurate SM for only a few in-situ SM stations. From Fig. S1 (b), we can see that the total numbers of data at 70 cm and 90 cm soil depths are quite small. In other words, more abundant of data were expected to help RF model 'learn' complete relationship between covariates and in-situ SM and further improve the quality of high-resolution SM in China.    Meanwhile, compared with the previous study of Sungmin et al. (2020), our SMCI1.0 showed the superior quality (Figure 4-6), because the larger numbers of in-situ SM data in China were applied for RF modelling.

From Figure 5, during the rainfall near 91th day across the Tropical Monsoon Climate zone (Am) and near 1st day across the Snow climate with dry winter zone (Dw), the in-situ SM did not increase

with high precipitation, but the SMCI1.0 product could capture the increase in SM (denoted in the light blue rectangle). The reason may be that the applied covariates had bias with in-situ measurement and further affected estimation by RF model. Meanwhile, we also found the RF model could overcome much bias in dry conditions, except for that from 196th to 305th days in the snow climate, fully humid zone (shown in the light red rectangle). In the case of 30 cm soil depth (Fig. S5), we could see an agreement between several peak events, it could be attributed to the soil texture homogeneity at the 10 and 30 cm soil depths. Almost all climatic regions had lower dynamic ranges at 30 cm soil depth than that at 10 cm, this may be attributed to the persistent behaviour of SM at 30 cm soil depth. In the case of 30 cm soil depth in Fig. S6, the SMCI1.0 product had higher accuracy than that at 10 cm soil depth (Figure 6), especially in terms of ubRMSE and MAE metrics. The reason may be the background aridity led to low variability of SM in the deeper layers (Karthikeyan and Mishra 2021) and the RF model can capture the variation in SM easier.

Interestingly, it was inconsistent for the results of R, ubRMSE, and MAE in Fig. 2 and Fig. 4, which had the same phenomenon with the previous study (Sungmin and Orth 2020) (represented in their Fig. 4 and Fig. 5). For example, SMCI1.0 product had the ubRMSE, MAE and R being 0.046, 0.035 and 0.889 at 10 cm soil depth in Fig. 2. However, in Fig. 4, the box-plot represented the lowest ubRMSE, MAE and highest R of SMCI1.0 product were nearly 0.03, 0.02, and 0.7, respectively. The reason may be that the same metrics were calculated in different ways, the one in Fig. 2 was to count the results of all stations and temporal period, and the one in Fig. 4 was to count the results of only temporal period at one station.

It was necessary to note that we also compared the RF model with other ML models, including CatBoost (Dorogush et al. 2018), XgBoost (Chen et al. 2016), and Neural Network (Rosenblatt et al. 1958) based models. We found that the performance of these models is very similar to RF models with a R2 around 0.79.   In addition, RF has been widely applied and recognized in SM prediction and many other fields (Carranza et al. 2021, Lin et al. 2022, Ly et al. 2021) and it does not take too much computing time to make the predictions for the whole China. Hence, we only took RF model to produce the high-resolution SM data.

Comment#5: The "Discussion" really needs to be reorganized and improved; the current one does not provide deep thoughts on the new soil moisture products, in terms of their differences/similarities/uniqueness compared to previous products/work and implications for the soil moisture modeling and detection and attribution.

Responds:

Thanks for your kind comments and helpful suggestions, we first removed the Section 4.1 in the old manuscript and put the related text into the "Conclusions" part. The new expression is as follows:
In this study, the gridded soil moisture was estimated through RF method in China based on the ERA5-Land reanalysis, USGS land cover type and DEM, reprocessed LAI and soil properties from CSDL, which included soil depths from 10cm to 100cm and had 1km spatial and daily temporal resolution over the period from 1 January 2010 to 31 December 2020.

We set "Sensitivity to precipitation, air temperature and radiation" as Section 4.2, as it is close to the new Section 4.1. We set Section 4.3 as "Factors affecting the quality of SMCI1.0". We combined the original Section 4.3 and 4.4 as the new Section 4.4 "Requirement of further validations and

improvements".

In addition, we have added the Section 4.5 providing some thoughts on our product about implications for the soil moisture modeling and attribution, meanwhile, in this section, we have also added the discussion about comparison between our product and previous products. The new expression is as follows:

In this section, we mainly discussed the comparison between SMCI1.0 and previous products, and the implications for the soil moisture modeling and attribution. From the previous results in Section 3, we can see that SMCI1.0 generally outperforms the existing SM products (ERA5-Land, SoMo.ml and SMAP-L4) at most cases. The most important uniqueness of SMCI1.0 is taking the in-situ SM data as the training target with abundant sample size. Even though we used the ERA5-land to correct their means and standard deviation at each site, the temporal variation still came from the observations. We have also tested to train the RF model with the original SM observations and found that the performance of the model decreased dramatically with a R2 of 0.67 compared to the model with correction (a R2 of 0.79). And more importantly, the resulting SM maps demonstrated unreasonable noisy spatial distribution. These indicates that the in-situ SM in China have essential data inconsistency and the correction according to ERA5-Land is necessary which has physical consistency. Furthermore, SMCI1.0 is provided with relatively high spatial and temporal resolution (1-km and daily) for ten soil depths, which makes it possible for wider applications at finer scales and deep soils for the whole China, while reanalysis and remote sensing SM data are often at coarser resolution and remote sensing SM data are only for the surface soil.

However, SMCI1.0 estimated by machine learning model cannot always reflect the variation of SM well, especially for some extreme events or so called "tipping points" (Bury et al. 2021). From Fig,5, we can see that SMCI1.0 deviated from the in situ SM in some cases, though this also happed to the other three SM products. For example, from 35th day to 61th day across the Snow climate, fully humid (Df), SMCI1.0 and SoMo.ml overestimated, while SMAP_L4 underestimated. "Tipping points" denoted that slowly changing SM sparks a sudden shift to a new (Bury et al. 2021). This is a huge challenge for estimating in-situ SM by ML models, because "tipping points" make the dynamics of complex system simplify down to the limited number of possible "normal forms" (Bury et al. 2021). ML models cannot accurately capture such extreme events. Hence, for these extreme events, we hope ML models trained on a sufficiently diverse database of possible SM variation, so that complex relationship between SM and predictors will be captured better and "tipping points" will be approached. In the future work, a possible solution is to apply a Land surface model, such as Common Land Model (Dai et al. 2003), to simulate large numbers of SM data and select the local bifurcations in SM variation as supplementary samples.

Comment#6: Grammar mistakes can be noticed in many places, for example, for the sentences between lines 82-87, 91-06, and 112-114 among others. The authors are suggested to get help from native English speakers and thoroughly check the whole manuscript before the next submission.

Sorry for the grammar mistakes. We have carefully checked the whole manuscript and revised the inaccurate description. We will also ask a native English speaker to help us for English revision before the next submission.

---

## Author Response (AR1)

Note: The modifications are shown in green. The responses to comments are blue colored.

**Response to RC1**

We are very grateful to the reviewer for reviewing the paper so carefully. These comments are very helpful to improve the quality of the manuscript. Please find our itemized responses below and our revisions will be in the revised manuscript.

Comment#1: In this study, only the Random Forest method was applied to derive the upscaled soil moisture data. Is it possible to try more MLs, such as CatBoost, XgBoost, and NeuralNetwork, to test how consistent or different the resulted products are?

Responds:

According to previous studies introduced in the introduction, RF models have proved to be successful in producing SM data and its computation time is acceptable. So, we choose this model in our studies.

According to this comment, we have also tested three more ML models (CatBoost, XgBoost, and Multilayer Perception) and found that the performance of these models is very close to RF. We applied the *scikit-learn* tools to evaluate them. We optimized the model parameters as follows. For the CatBoost model, we used the default parameters in catboost library. For the XgBoost model, we optimize the XgBoost by tuning the parameters through the *selection.GridSearchCV* function which is provided in the *scikit-learn tools*, we set *learning_rate* being 5.0e-3, *max_depth* being 6, *randam_state* being 42, and *estimators* being 100. For the Multilayer Perception, we set *random_state* being 1 and *max_iter* being 500. Take the year-to-year experiment at 10 cm soil depth as an example (see the figure below), RF achieved *ubRMSE* being 0.046, *MAE* being 0.035, *R* being 0.889, $R^2$ being 0.791. The *ubRMSE, MAE, R, $R^2$* of CatBoost were 0.045, 0.034, 0.890 and 0.799, respectively. XgBoost achieved *ubRMSE* being 0.045, *MAE* being 0.034, *R* being 0.890, $R^2$ being 0.799. The *ubRMSE, MAE, R, $R^2$* of ANN-based model were 0.045, 0.035, 0.891 and 0.796, respectively. According to these results, we don't think it is necessary to use other models instead of RF to produce the high-resolution SM as they cann't outperform it much.

[Figure]

In the modified manuscript, we have added the explanation why only RF model was shown in the

discussion. The new expression is as follows:

The obtained results by RF method were also compared with those of some other ML models, including CatBoost (Dorogush et al. 2018), XgBoost (Chen et al. 2016), and Neural Network (Rosenblatt et al. 1958) models. We found that their performance is similar to RF models with a R2 value around 0.79.   Therefore, due to the comparable performance and wide application of RF to SM modelling (e.g., Carranza et al. 2021, Lin et al. 2022, Ly et al. 2021), and more importantly due to its cost-effective run time, only the results of RF were considered to produce high-resolution SM data in this study.

Comment#2: Most of the source datasets cover the period before year 2010. Is there any special reason why the new soil moisture only covers the period 2010-2020? Is it possible to extend the present time period to year 2000-2020?

Responds:

Thanks for your kind comments and helpful suggestions. Although most of the applied covariates cover the period before year 2010, we do not access to the *in-situ* measurements before 2010, currently. The in-situ measurements before 2010 may be available from China Meteorological Administration (not open to us) and the number of stations is less than 800. If we produce the SM data set without any in-situ data (or only a few hundred stations), the quality of the data may be poorer as it will be extrapolation in time. However, we agree that it is proper (assuming the relationship between SM and covariates remains the same in the last two decades) to extend the present time period to 2000-2020. We did not extent it before 2000 taking a conservative attitude. But it is possible to extend it as long as in-situ SM is available in the future.   The extended data is still available at http://dx.doi.org/10.11888/Terre.tpdc.272415.   We have added the following contents in section 2.4:

In addition to the period of 2010-2020 when in situ SM data are available, we also produced the gridded SM for the period of 2000-2009 when in situ SM data are unavailable, assuming that the relationship between SM and predictors remains the same in the last two decades. It is proper to deem that the data quality during 2000-2009 is poorer than that of 2010-2020.

We also list a future work in the conclusion as follows:

It is also possible to update and extent the time coverage of this data set before 2010 as long as in situ SM data becomes available.

Comment#3: The "Materials and Methods" read too long, and the authors may try to shorten the text and put some figures into the Supplementary Material.

Responds:

According to this comment, we have put the Figure 2 and related text into the Supplementary Material. We also shortened the text in this section as follows.

In section 2.1, we combined the following text with the last paragraph of this section:

"The other aspect is the bias and standard deviation correction of in-situ SM, which is vital for our

study to allow the ML model to achieve the high-quality SM product. We applied the same correcting method with that of Sungmin et al. (2020), who adjusted the raw in-situ SM observations to match means and standard deviation of the ERA5-Land gridded SM data at the corresponding time periods, grid cells and layers." (deleted)

The last paragraph of this section:

After the above data processing, we started to perform the correction of deviation and variance for in-situ SM, which is vital for our study to allow the ML model to achieve the high-quality SM product. In-situ SM data was obtained by various sensor types, which had different calibrations. Hence, to overcome the artifacts during the RF model training, we adjusted the observations to match means and standard deviation of the ERA5-Land SM at the corresponding time periods, grid cells and layers using the same method with that of Sungmin and Orth (2020). This method made the target in-situ SM resemble the mean and standard deviation of ERA5-Land SM, and kept daily temporal variations which follow the original in-situ SM time series. As the soil depth of each soil layer of ERA5-Land SM was inconsistent with that of in-situ SM, we mapped the soil layer of ERA5-Land SM to the corresponding soil layers of in-situ SM. Hence, the in-situ SM from 10 cm to 30 cm were adjusted based on the gridded SM at layer2 from ERA5-Land dataset (7-28 cm), and the in-situ SM from 30 cm to 100 cm were adjusted based on the gridded SM at layer3 from ERA5-Land dataset (28-100 cm).

We have also put the Table 2 into the Supplementary Material and shortened the related paragraph as follows:

After the above data processing, the correction of deviation and variance of in-situ SM was performed, which can help the ML model to achieve the high-quality SM product. In-situ SM data have been obtained by various sensor types with different calibration processes. Hence, to overcome the artefacts during the RF model training, we adjusted the observations to match means and standard deviation of the ERA5-Land SM at the corresponding time periods, grid cells and layers using the method proposed by Sungmin and Orth (2020). In this method, we first obtained a weight by dividing the standard deviations of the in-situ SM at each station by that of ERA5-Land SM at the corresponding grid, and then multiplied the original in-situ SM by this weight. After that, we computed the difference between the average value of the in-situ SM at each station and the ERA5-Land SM at the corresponding grid, and subtract the in-situ SM by the computed difference. This method made the target in-situ SM resemble the mean and standard deviation of ERA5-Land SM, and kept daily temporal variations which follow the original in-situ SM time series. As the soil depth of each soil layer of ERA5-Land SM was inconsistent with that of in-situ SM, we mapped the soil layer of ERA5-Land SM to the corresponding soil layers of in-situ SM. Hence, the in-situ SM data from 10 cm to 30 cm were adjusted based on the gridded SM at layer2 from ERA5-Land dataset (7-28 cm), and the in-situ SM data from 30 cm to 100 cm were adjusted based on the gridded SM at layer3 from ERA5-Land dataset (28-100 cm).

We also revised shortened other contents. This has been shown in the revised manuscript with track-changes.

Comment#4: For the "Results", they seem to be a combination of results analysis and short

discussion. Please move relevant discussion content to the "Discussion" part.

Responds:

Thanks for your kind comments and helpful suggestions, we have moved Section 3.4 in the old manuscript to the discussion as Section 4.1, and put some short discussions in the results of the old manuscript into the "Discussion" part. The new section in the discussion is as follows:

Fig. 2 and S2 show that SM results at 70 cm and 90 cm were significant worse than those at other depths. The reason may be that linked to the incapability of the RF model to estimate accurate SM when data from only a few in-situ SM stations are available. From Fig. S1 (b), we can see that the total numbers of data at 70 cm and 90 cm soil depths are quite small. In other words, more abundant of data could help RF model to 'learn' relationship between predictors and in-situ SM data reliably and further improve the quality of high-resolution SM estimation over China. Meanwhile, compared to the previous study of Sungmin et al. (2020), our SMCI1.0 showed the superior quality (Fig. 4-6), because the larger numbers of in-situ SM data of China wereapplied for the RF based modelling.

From Fig. 5, during the rainfall near 91th day across the Tropical Monsoon Climate zone (Am) and near 1st day across the Snow climate with dry winter zone (Dw), the in-situ SM values did not increase due to high precipitation, but the SMCI1.0 product could capture the increase in SM (denoted in the light blue rectangle). The reason may be that the applied predictors had bias with in-situ measurements and further affected the SM estimation by RF model. Meanwhile, we also found the RF model could overcome much bias in dry conditions, except for those from 196th to 305th days in the snow climate, fully humid zone (shown in the light red rectangle). In the case of 30 cm soil depth (Fig. S5), we could see an agreement between several peak events, it could be attributed to the soil texture homogeneity at the 10 and 30 cm soil depths. Almost all climatic regions had lower dynamic ranges at 30 cm soil depth than that at 10 cm, this may be attributed to the persistent behaviour of SM at 30 cm soil depth. In the case of 30 cm soil depth in Fig. S6, the SMCI1.0 product had higher accuracy than that at 10 cm soil depth (Fig. 6), especially in terms of ubRMSE and MAE metrics. The reason may be due to the background aridity which could lead to low variability of SM in the deeper layers (Karthikeyan and Mishra 2021) so that the RF model could capture the SM variation in SM straightforwardly.

Oppositely, it is inconsistent for the results of R, ubRMSE, and MAE in Fig. 2 and Fig. 4, which is similar to the previous study (Sungmin and Orth 2020) (represented in their Fig. 4 and Fig. 5). For example, SMCI1.0 product led to the ubRMSE, MAE and R values being 0.046, 0.035 and 0.889 at 10 cm soil depth in Fig. 2. However, in Fig. 4, the box-plot shows the lowest ubRMSE, MAE and highest R values of SMCI1.0 product as 0.03, 0.02, and 0.7, respectively. The reason may be due to the circumstances of computing the same metrics in different ways, so that the results of Fig. 2 are for all stations and temporal period, whereas Fig. 4 shows the results of temporal period at only one station.

The obtained results by RF method were also compared with those of some other ML models, including CatBoost (Dorogush et al. 2018), XgBoost (Chen et al. 2016), and Neural Network (Rosenblatt et al. 1958) models. We found that their performance is similar to RF models with a R2 value around 0.79. Therefore, due to the comparable performance and wide application of RF to SM modelling (e.g., Carranza et al. 2021, Lin et al. 2022, Ly et al. 2021), and more importantly due

to its cost-effective run time, only the results of RF were considered to produce high-resolution SM data in this study.

Comment#5: The "Discussion" really needs to be reorganized and improved; the current one does not provide deep thoughts on the new soil moisture products, in terms of their differences/similarities/uniqueness compared to previous products/work and implications for the soil moisture modeling and detection and attribution.

Responds:

Thanks for your kind comments and helpful suggestions, we first removed the Section 4.1 in the old manuscript and put the related text into the "Conclusions" part. The new expression is as follows:

In this study, the gridded SM data was estimated through RF method over China based on the ERA5-Land reanalysis, USGS land cover type and DEM, reprocessed LAI and soil properties from CSDL, which included soil depths from 10cm to 100cm and had 1km spatial and daily temporal resolution over the period from 1 January 2010 to 31 December 2020.

We set "Sensitivity to precipitation, air temperature and radiation" as Section 4.2, as it is close to the new Section 4.1. We set Section 4.3 as "Factors affecting the quality of SMCI1.0". We combined the original Section 4.3 and 4.4 as the new Section 4.4 "Requirement of further validations and improvements".

In addition, we have added the Section 4.5 providing some thoughts on our product about implications for the soil moisture modeling and attribution, meanwhile, in this section, we have also added the discussion about comparison between our product and previous products. The new expression is as follows:

This section mainly described and discussed the comparison between SMCI1.0 and some other SM products, and the implications for the soil moisture modelling and attribution. From the results presented in Section 3, we can see that SMCI1.0 generally outperforms some other SM products (e.g., ERA5-Land, SoMo.ml and SMAP-L4) at most cases. The most important uniqueness of SMCI1.0 is taking the in-situ SM data as the training target with abundant sample size. Even though we used the ERA5-Land to correct their means and standard deviation at each site, the temporal variation still came from the point observations. We have also examined the RF model training with the original SM observations and found that the performance of the model is much worse with a R2 of 0.67 compared to the model with correction with a R2 of 0.79. More importantly, the resulting SM maps demonstrated unreasonable noisy spatial distribution. These indicates that the in-situ SM in China have essential data inconsistency and the correction according to ERA5-Land is necessary which has physical consistency. Furthermore, SMCI1.0 has been provided with relatively high spatial and temporal resolutions (1-km, daily) for ten soil depths, which makes it possible for wider applications at finer scales and deep soils for the whole China, while reanalysis and remote sensing SM data are often at coarser resolution and remote sensing SM data are only for the surface soil.

As the limitation for the SMCI1.0, machine learning based model cannot always reflect the variation of SM well, especially for some extreme events or so called "tipping points" (Bury et al. 2021). From Fig,5, we can see that SMCI1.0 deviated from the in-situ SM data in some cases, though this also happened to the other three SM products. For example, from 35th day to 61th day across the Snow climate, fully humid (Df), SMCI1.0 and SoMo.ml overestimated, while SMAP_L4

underestimated. "Tipping points" denoted that slowly changing SM sparks a sudden shift to a new (Bury et al. 2021). This discontinuity creates a big challenge for estimating in-situ SM by ML models, because "tipping points" simplify the dynamics of complex system down to the limited number of possible "normal forms" (Bury et al. 2021). ML models cannot accurately capture such extreme events. Hence, for these extreme events, we hope ML models trained on a sufficiently diverse datasets of possible SM variation can well capture complex relationship between SM and predictors. As a suggestion for the future work, a possible solution for this limitation is to apply a Land surface model, such as Common Land Model (Dai et al. 2003), to simulate large numbers of SM data and select the local bifurcations in SM variation as supplementary samples to enhance the learning generality of the RF model.

Comment#6: Grammar mistakes can be noticed in many places, for example, for the sentences between lines 82-87, 91-06, and 112-114 among others. The authors are suggested to get help from native English speakers and thoroughly check the whole manuscript before the next submission.

Sorry for the grammar mistakes. We have carefully checked the whole manuscript and revised the inaccurate description. We asked a native English speaker to help us for English revision. Please see the track-changes.

**Response to RC2**

Note: The modifications are shown in green. The responses to comments are blue colored.
We are very grateful to Reviewer for reviewing the paper so carefully. These comments are very helpful to improve the quality of the manuscript. Please find my itemized responses in below and my revisions will be in the re-submitted files.

**Major comments:**

Comment#1: The paper writing is extremely poor. Language errors and statement repeats or inconsistency are found through the whole manuscript. For example, "dataset of China" in the title should be "dataset over China" or "dataset for China"; in Lines 14-15, the authors have stated "high quality gridded soil moisture products" are "usually available from remote sensing… with coarse resolution" but then they raise that "high quality" is characterized by "high-resolution…", which is obviously contradictory; there are many other cases like the usage of "… is acted as".

Responds:We corrected the errors mentioned by the reviewer. We have also carefully checked the whole manuscript and revised the inaccurate description. We also invited a native English speaker to polish the manuscript. Please see the track-changes. The sentence in lines 14-15 is corrected as: High quality gridded soil moisture products are essential for many Earth system science applications, while the recent reanalysis and remote sensing soil moisture data are often available at coarse resolution and remote sensing data are only for the surface soil.

Comment#2: More importantly, the Results part contains many statements that are actually discussion while the Discussion parts contains too many results.

Responds:Thanks for your kind comments and helpful suggestions. We have moved Section 3.4 in the old manuscript to the discussion as Section 4.1, and put some short discussions in the results of the old manuscript into the "Discussion" part. The new section in the discussion is as follows:
4.3 Factors affecting the quality of SMCI1.0
Fig. 2 and S2 show that SM results at 70 cm and 90 cm were significant worse than those at other depths. The reason may be that linked to the incapability of the RF model to estimate accurate SM when data from only a few in-situ SM stations are available. From Fig. S1 (b), we can see that the total numbers of data at 70 cm and 90 cm soil depths are quite small. In other words, more abundant of data could help RF model to 'learn' relationship between predictors and in-situ SM data reliably and further improve the quality of high-resolution SM estimation over China. Meanwhile, compared to the previous study of Sungmin et al. (2020), our SMCI1.0 showed the superior quality (Fig. 4-6), because the larger numbers of in-situ SM data of China wereapplied for the RF based modelling.
From Fig. 5, during the rainfall near 91th day across the Tropical Monsoon Climate zone (Am) and near 1st day across the Snow climate with dry winter zone (Dw), the in-situ SM values did not increase due to high precipitation, but the SMCI1.0 product could capture the increase in SM

(denoted in the light blue rectangle). The reason may be that the applied predictors had bias with in-situ measurements and further affected the SM estimation by RF model. Meanwhile, we also found the RF model could overcome much bias in dry conditions, except for those from 196th to 305th days in the snow climate, fully humid zone (shown in the light red rectangle). In the case of 30 cm soil depth (Fig. S5), we could see an agreement between several peak events, it could be attributed to the soil texture homogeneity at the 10 and 30 cm soil depths. Almost all climatic regions had lower dynamic ranges at 30 cm soil depth than that at 10 cm, this may be attributed to the persistent behaviour of SM at 30 cm soil depth. In the case of 30 cm soil depth in Fig. S6, the SMCI1.0 product had higher accuracy than that at 10 cm soil depth (Fig. 6), especially in terms of ubRMSE and MAE metrics. The reason may be due to the background aridity which could lead to low variability of SM in the deeper layers (Karthikeyan and Mishra 2021) so that the RF model could capture the SM variation in SM straightforwardly.

Oppositely, it is inconsistent for the results of R, ubRMSE, and MAE in Fig. 2 and Fig. 4, which is similar to the previous study (Sungmin and Orth 2020) (represented in their Fig. 4 and Fig. 5). For example, SMCI1.0 product led to the ubRMSE, MAE and R values being 0.046, 0.035 and 0.889 at 10 cm soil depth in Fig. 2. However, in Fig. 4, the box-plot shows the lowest ubRMSE, MAE and highest R values of SMCI1.0 product as 0.03, 0.02, and 0.7, respectively. The reason may be due to the circumstances of computing the same metrics in different ways, so that the results of Fig. 2 are for all stations and temporal period, whereas Fig. 4 shows the results of temporal period at only one station.

The obtained results by RF method were also compared with those of some other ML models, including CatBoost (Dorogush et al. 2018), XgBoost (Chen et al. 2016), and Neural Network (Rosenblatt et al. 1958) models. We found that their performance is similar to RF models with a R2 value around 0.79. Therefore, due to the comparable performance and wide application of RF to SM modelling (e.g., Carranza et al. 2021, Lin et al. 2022, Ly et al. 2021), and more importantly due to its cost-effective run time, only the results of RF were considered to produce high-resolution SM data in this study.

We also removed the Section 4.1 in the old manuscript and put the related text into the "Conclusions" part. The new expression is as follows:

In this study, the gridded SM data was estimated through RF method over China based on the ERA5-Land reanalysis, USGS land cover type and DEM, reprocessed LAI and soil properties from CSDL, which included soil depths from 10cm to 100cm and had 1km spatial and daily temporal resolution over the period from 1 January 2010 to 31 December 2020.

Finally, we set "Sensitivity to precipitation, air temperature and radiation" as Section 4.2, as it is close to the new Section 4.1. We set Section 4.3 as "Factors affecting the quality of SMCI1.0". We combined the original Section 4.3 and 4.4 as the new Section 4.4 "Requirement of further validations and improvements". In addition, we have added the Section 4.5 providing some thoughts on our product about implications for the soil moisture modeling and attribution, meanwhile, in this section, we have also added the discussion about comparison between our product and previous products. The new expression is as follows:

This section mainly described and discussed the comparison between SMCI1.0 and some other SM products, and the implications for the soil moisture modelling and attribution. From the results presented in Section 3, we can see that SMCI1.0 generally outperforms some other SM products (e.g., ERA5-Land, SoMo.ml and SMAP-L4) at most cases. The most important uniqueness of

SMCI1.0 is taking the in-situ SM data as the training target with abundant sample size. Even though we used the ERA5-Land to correct their means and standard deviation at each site, the temporal variation still came from the point observations. We have also examined the RF model training with the original SM observations and found that the performance of the model is much worse with a R2 of 0.67 compared to the model with correction with a R2 of 0.79. More importantly, the resulting SM maps demonstrated unreasonable noisy spatial distribution. These indicates that the in-situ SM in China have essential data inconsistency and the correction according to ERA5-Land is necessary which has physical consistency. Furthermore, SMCI1.0 has been provided with relatively high spatial and temporal resolutions (1-km, daily) for ten soil depths, which makes it possible for wider applications at finer scales and deep soils for the whole China, while reanalysis and remote sensing SM data are often at coarser resolution and remote sensing SM data are only for the surface soil.

As the limitation for the SMCI1.0, machine learning based model cannot always reflect the variation of SM well, especially for some extreme events or so called "tipping points" (Bury et al. 2021). From Fig,5, we can see that SMCI1.0 deviated from the in-situ SM data in some cases, though this also happened to the other three SM products. For example, from 35th day to 61th day across the Snow climate, fully humid (Df), SMCI1.0 and SoMo.ml overestimated, while SMAP_L4 underestimated. "Tipping points" denoted that slowly changing SM sparks a sudden shift to a new (Bury et al. 2021). This discontinuity creates a big challenge for estimating in-situ SM by ML models, because "tipping points" simplify the dynamics of complex system down to the limited number of possible "normal forms" (Bury et al. 2021). ML models cannot accurately capture such extreme events. Hence, for these extreme events, we hope ML models trained on a sufficiently diverse datasets of possible SM variation can well capture complex relationship between SM and predictors. As a suggestion for the future work, a possible solution for this limitation is to apply a Land surface model, such as Common Land Model (Dai et al. 2003), to simulate large numbers of SM data and select the local bifurcations in SM variation as supplementary samples to enhance the learning generality of the RF model.

Comment#3: Additionally, much discussion in the Results and Discussion sections actually lacks sufficient evidence support (e.g., Lines 361-362).

Responds: We have carefully checked the whole manuscript and deleted the inappropriate discussions including line 361-362.

Comment#4: The soil moisture product has a spatial resolution of 1 km while the input data, ERA5-Land product, has a resolution of 9 km. So how did the `authors pre-reprocess the ERA5-Land data?

Responds: Thanks for your kind comments and helpful suggestions. We have described the pre-reprocessing of the ERA5-Land data as follows:
All predictors were processed to the same 1km by 1km grid system. ERA5-Land data with 9 km resolution were resampled into 1 km by the nearest neighbor method and MODIS LAI with 500 m resolution were aggregated into 1 km by averaging.

Comment#5: They mentioned that in-situ observations were adjusted to ERA5-Land soil moisture

but did not introduce the specific methodology.

Responds:For adjusting the in-situ observations to ERA5-Land soil moisture, we have added the specific methodology as follows:

.

In this method, we first obtained a weight by dividing the standard deviations of the in-situ SM at each station by that of ERA5-Land SM at the corresponding grid, and then multiplied the original in-situ SM by this weight. After that, we computed the difference between the average value of the in-situ SM at each station and the ERA5-Land SM at the corresponding grid, and subtract the in-situ SM by the computed difference. This method made the target in-situ SM resemble the mean and standard deviation of ERA5-Land SM, and kept daily temporal variations which follow the original in-situ SM time series.).

**Minor comments:**

Comment#6: The soil moisture product ranges from 2010-2020 but this time coverage is still too short for analysis in related fields, for example, the occurrence of droughts. I am wondering why the authors chose such a target period.

Responds:Thanks for your kind comments and helpful suggestions. The in-situ measurements before 2010 may be available from China Meteorological Administration (not open to us) and the number of stations is less than 800. If we produce the SM data set without any in-situ data (or only a few hundred stations), the quality of the data may be poorer as it will be extrapolation in time. However, we agree that it is proper (assuming the relationship between SM and covariates remains the same in the last two decades) to extend the present time period to 2000-2020. We did not extent it before 2000 taking a conservative attitude. But it is possible to extend it as long as in-situ SM is available in the future. The extended data is still available at http://dx.doi.org/10.11888/Terre.tpdc.272415. We have added the following contents in section 2.4:

In addition to the period of 2010-2020 when in situ SM data are available, we also produced the gridded SM for the period of 2000-2009 when in situ SM data are unavailable, assuming that the relationship between SM and predictors remains the same in the last two decades. It is proper to deem that the data quality during 2000-2009 is poorer than that of 2010-2020.

We also list a future work in the conclusion as follows:

It is also possible to update and extent the time coverage of this data set before 2010 as long as in situ SM data becomes available.

Comment#7: In the text, the authors mentioned terms such as "Liaoning province", "Sichuan province" and "the plateau", which are not friendly to readers that have no the background knowledge.

Responds:Thanks for your kind comments and helpful suggestions, we have added the detailed

longitude and latitude to the mentioned regions. The new expression is as follows:

Meanwhile, SMCI1.0 product often underestimated in north China and overestimated in Sichuan province (97°21'E-108°12'E, 26°03'N-34°19'N)

Additionally, air temperature had significant positive partial correlations with SM in the northwestern China, and negative partial correlations in north China and Liaoning province (118°53'E-125°46'E, 38°43'N-43°26'N) for SMCI1.0.

Qinghai province (89°35'E-103°04'E, 31°09'N-39°19'N) belongs to the tundra climate zone, where some soils are wet and other soils are dry.

In some of the plateau areas (73°19'E-104°47'E, 26°00'N-39°47'N)

**Response to CC1**

Note: The modifications are shown in green. The responses to comments are blue colored.
We are very grateful to Reviewer for reviewing the paper so carefully. These comments are very helpful to improve the quality of the manuscript.

**Reviewer's comment:**

The citation for ERA5-Land is completely wrong. Instead, the authors cite an old dataset (ERA-Interim/Land), and even this citation is wrong as well. The right citation of ERA5-Land can be consulted here: https://confluence.ecmwf.int/display/CKB/ERA5-Land%3A+data+documentation#ERA5Land:datadocumentation-HowtocitetheERA5-Landdataset. In that link it is clearly indicated how to cite the dataset and how to acknowledge the authors as stated in the Copernicus C3S/CAMS License agreement.

Therefore, I would kindly request the authors to modify and rightly provide the ERA5-Land citation in the manuscript, but also check the rest of citations, which also contain lot of problems.

**Responds to the reviewers' comments:**

We thank Dr. Muñoz Sabater, the creator of ERA5-Land, for pointing this out. We are sorry for the wrong citation. We have revised the citation of ERA5-Land as following (format according to ESSD):

Muñoz Sabater, J.: ERA5-Land hourly data from 1981 to present. Copernicus Climate Change Service (C3S) Climate Data Store (CDS) [data set], https://doi.org/10.24381/cds.e2161bac, 2019.

Muñoz Sabater, J.: ERA5-Land hourly data from 1950 to 1980. Copernicus Climate Change Service (C3S) Climate Data Store (CDS) [data set], https://doi.org/10.24381/cds.e2161bac, 2021.

We will also check the rest of citations as the reviewer suggested.

**Response to CC2**

Note: The modifications are shown in green. The responses to comments are blue colored.
We are very grateful to Reviewer for reviewing the paper so carefully. These comments are very helpful to improve the quality of the manuscript. Please find my itemized responses in below and my revisions will be in the re-submitted files.

Comment#1: "Table 1 shows the datasets uses covariates used for RF modeling. Most covariates were collected from the ERA5-Land reanalysis dataset, which was produced by the land component of European Centre for Medium-Range Weather Forecasts (ECMWF)."
Wrong, ERA5-Land wasn't produced by the land component of ECMWF. What does it mean??? Instead, ERA5-Land is an enhanced version of the ERA5 land component.

Responds:

According to this comment and the description at https://confluence.ecmwf.int/display/CKB/ERA5-Land%3A+data+documentation, We have modified the sentence as follows:
which is an enhanced version of ERA5 land component, forced by meteorological fields from ERA5.

Comment#2:"The reasons for selecting the ERA5-Land dataset as preference were as follows: (1) it is generated under a single simulation of a land surface model using ERA5 reanalysis as the forcing data, but with a series of improvements making it more accurate for all types of land applications (Albergel et al. 2018);"
Completely wrong reference. This is stated in [Muñoz-Sabater et al., 2021], not in [Albergel et al., 2018].

Responds:

We have revised this as (Muñoz-Sabater et al., 2021) and add the following citation.
Muñoz-Sabater, J., Dutra, E., Agustí-Panareda, A., Albergel, C., Arduini, G., Balsamo, G., Boussetta, S., Choulga, M., Harrigan, S., Hersbach, H., Martens, B., Miralles, D. G., Piles, M., Rodríguez-Fernández, N. J., Zsoter, E., Buontempo, C., and Thépaut, J.-N.: ERA5-Land: A state-of-the-art global reanalysis dataset for land applications, Earth Syst. Sci. Data,13, 4349–4383, https://doi.org/10.5194/essd-13-4349-2021, 2021.

Comment#3: "(2) there are only several months latency for obtaining ERA5-Land datasets"
What? What do you mean for obtaining E5L datasets? No, ERA5-Land is currently updated with 2-3 months latency.

Responds: Sorry for the inaccurate description. Obtaining it costs some time to download. The new expression is as follows:
ERA5-Land is currently updated with 2-3 months latency.

Comment#4: "(3) the data is long-term (since 1981) and with seamless 175 spatial distribution and

multilayers"

Wrong, the data is available since 1950, not 1981.

Responds: This is corrected.

---

## Author Response (AR2)

Note: The modifications are shown in green. The responses to comments are blue colored.

We are very grateful to Reviewer for reviewing the paper so carefully. These comments are very helpful to improve the quality of the manuscript. Please find my itemized responses in below and my revisions will be in the re-submitted files.

**Major comments:**

Comment#1-1 Why use the ERA5-LAND dataset instead of weather station data correspondence with the CMA SM Observations? As far as I know, each soil moisture observation station of China Meteorological Administration will have a corresponding meteorological station.

Responds:Thanks for your kind comments and helpful suggestions. The reason using the ERA5-LAND dataset is that it is a relative accurate dataset for all types of land applications and it has seamless spatial distribution. The SM data with seamless spatial distribution can be generated by establishing the relationship between ERA5-LAND dataset and CMA SM Observations. However, the meteorological stations are too sparse to capture adequate spatial coverage, which cannot cover the whole China. Our data generating method follows the current research of most soil moisture products (Sungmin and Orth 2020, Zeng et al. 2019, Carranza et al. 2020, Karthikeyan and Mishra 2021), which use spatial continuous meteorological data as the predictors instead of individual meteorological stations. As the results shown in our work, this way can already provide satisfying performance in the data producing.

However, the valuable opinion of the reviewer also gives us a new inspiration. In the future work, we can first collect the meteorological data from the corresponding meteorological station. To achieve the seamless spatial distribution, we then apply spatial interpolation method to generate the meteorological data, or just use other existing gridded meteorological data. Finally, RF model is used to establish the relationship between the covariates and CMA SM Observations, and the SM data with seamless spatial distribution is further generated.

[1] Carranza, C., Nolet, C., Pezij, M., and van der Ploeg, M.: Root zone soil moisture estimation with Random Forest, Journal of Hydrology, 593, 125840, https://doi.org/10.1016/j.jhydrol.2020.125840, 2021.
[2] Karthikeyan, L. and Mishra, A. K.: Multi-layer high-resolution soil moisture estimation using machine learning over the United States, Remote Sensing of Environment, 266, 112706, https://doi.org/10.1016/j.rse.2021.112706, 2021.
[3] O, S. and Orth, R.: Global soil moisture data derived through machine learning trained with in-situ measurements, Scientific Data, 8, 170, https://doi.org/10.1038/s41597-021-00964-1, 2021.
[4] Zeng, L., Hu, S., Xiang, D., Zhang, X., Li, D., Li, L., and Zhang, T.: Multilayer Soil Moisture Mapping at a Regional Scale from Multisource Data via a Machine Learning Method, Remote Sensing, 11, https://doi.org/10.3390/rs11030284, 2019.

Comment#1-2: Most of the study areas are not natural areas, but farmland areas that have been affected by agricultural management practices for a long time. However, the random forest model does not consider human management measures, such as fertilization, irrigation, farming measures, etc., which I think is the biggest deviation of the results of this study.

Responds:Thanks for your kind comments. The SM in farmland areas is indeed affected by more factors than that in natural areas. However, the human management measure data are hardly obtained accurately and not available covering the whole farmland areas. We agree that this kind of data will help in improving the SM estimation in crop land as long as they are available. However, we also argue that the SM observations have already contains the effect of human management to some extent, as these measures will directly affect SM itself.

Of course, according to the valuable comment, it also provides an important direction for our future study. Specifically, for a specific farmland area, we will first collect or even developing the agricultural management data, such as fertilization, irrigation, farming measures, etc., and further explore how the agricultural management data affect the variation in SM. It is expected to be an important supplementary scheme to optimize the SMCI1.0 product.

Revise:

Other predictors should also be explored to improve the SM prediction. For the farmland areas, human management measures such as fertilization and irrigation should be considered in a proper way, even though this kind of data are rarely available in a spatial continuous way for the whole China. Lack of the consideration about agricultural management practices in the SMCI1.0 may lead to some deviation in SM estimation of the crop land.

Comment#1-3: As far as I am concerned, the study area includes the Qinghai-Tibet Plateau and the Northeast Alpine region, which are typical areas of freeze-thaw soil. Is it compared with the results of soil moisture research in these regions in China? Such as "A first assessment of satellite and reanalysis estimates of surface and root-zone soil moisture over the permafrost region of Qinghai-Tibet Plateau" and others.

Responds:Thanks for your kind comments and helpful suggestions. We have added the assessment of SM maps from different products over the Qinghai-Tibet Plateau and the Northeast Alpine regions

Revise:

We also compared the SM estimation of the Qinghai-Tibet Plateau (74°00'E-104°00'E, 25°00'N-40°00'N) and the Northeast Alpine region (128°50'E-129°50'E, 45°50'N-46°70'N), as they are typical areas of freeze-thaw soil (Fig. S8 and S9). According to the previous study of Xing et al. (2021), the ERA5-Land often be overestimated compared to in-situ SM (see their Fig. 5). In Fig. S8, the SMCI1.0 SM is underestimated compared to ERA5-Land over Qinghai-Tibet Plateau, which is more closed to in-situ SM. Additionally, over both Qinghai-Tibet Plateau and Northeast regions (128°50'E-129°50'E, 45°50'N-46°70'N), the more details of SM spatial patterns for SMCI1.0 SM can be found than that of ERA5-land, SoMo.ml and SMAP-L4 SM (Fig. S8 and Fig. S9).

[Figure]

**Figure S8. Soil moisture maps from different products on 1st January 2016 over Qinghai-Tibet Plateau region. The resolution is 1km for SMC1.0, 9km for ERA5-land and SMAP-L4 and 0.25 degree for SoMo.ml.**

[Figure]

**Figure S9. Soil moisture maps from different products on 1st January 2016 over Northeast region. The resolution is 1km for SMC1.0, 9km for ERA5-land and SMAP-L4 and 0.25 degree for SoMo.ml.**